



Modeling canopy-induced turbulence in the Earth system: a unified parameterization of turbulent
exchange within plant canopies and the roughness sublayer (CLM-ml v0)
Gordon B. Bonan[1]
Edward G. Patton[1]
Ian N. Harman[2]
Keith W. Oleson[1]
John J. Finnigan[2]
Yaqiong Lu[1]
Elizabeth A. Burakowski[3]
1 National Center for Atmospheric Research, P. O. Box 3000, Boulder, Colorado, USA 80307
2 CSIRO Oceans and Atmosphere, Canberra, Australia
3 University of New Hampshire, Durham, New Hampshire, USA
Corresponding author: G. B. Bonan (bonan@ucar.edu)




**Abstract**. Land surface models used in climate models neglect the roughness sublayer and
parameterize within-canopy turbulence in an ad hoc manner. We implemented a roughness
sublayer turbulence parameterization in a multi-layer canopy model (CLM-ml v0) test if this
theory provides a tractable parameterization extending from the ground through the canopy and
the roughness sublayer. We compared the canopy model with the Community Land Model
(CLM4.5) at 7 forest, 2 grassland, and 3 cropland AmeriFlux sites over a range of canopy height,
leaf area index, and climate. The CLM4.5 has pronounced biases during summer months at
forest sites in mid-day latent heat flux, sensible heat flux, and gross primary production,
nighttime friction velocity, and the radiative temperature diurnal range. The new canopy model
reduces these biases by introducing new physics. The signature of the roughness sublayer is most
evident in sensible heat flux, friction velocity, and the diurnal cycle of radiative temperature.
Within-canopy temperature profiles are markedly different compared with profiles obtained
using Monin–Obukhov similarity theory, and the roughness sublayer produces cooler daytime
and warmer nighttime temperatures. The herbaceous sites also show model improvements, but
the improvements are related less systematically to the roughness sublayer parameterization in
these short canopies. The multi-layer canopy with the roughness sublayer turbulence improves
simulations compared with the CLM4.5 while also advancing the theoretical basis for surface
flux parameterizations.
Keywords: multi-layer canopy, roughness sublayer, Monin–Obukhov similarity theory, wind
profile, scalar profile, land surface model





## 1 Introduction

Distinct parameterizations of land surface processes, separate from the atmospheric physics,
were coupled to global climate models in the mid-1980s with the Biosphere–Atmosphere
Transfer Scheme (BATS; Dickinson et al., 1986) and the Simple Biosphere Model (SiB; Sellers
et al., 1986). While carbon cycle feedbacks have since gained prominence in terms of model
development and study of biotic feedbacks with climate change (Friedlingstein et al., 2006,
2014), the fundamental coupling between plants and the atmosphere in climate models still
occurs with the fluxes of momentum, energy, and mass over the diurnal cycle as mediated by
plant physiology, the microclimate of plant canopies, and boundary layer processes. The central
paradigm of land surface models, as originally devised by Deardorff (1978) and carried forth
with BATS, SiB, and subsequent models, has been to represent plant canopies as a homogeneous
"big leaf" without vertical structure, though with separate source fluxes for vegetation and soil.
A critical advancement was to analytically integrate leaf physiological processes over profiles of
light and nitrogen in the canopy (Sellers et al., 1996) and to extend the canopy to two big leaves
to represent sunlit and shaded portions of the canopy (Wang and Leuning, 1998; Dai et al.,

2004).

In land surface models such as the Community Land Model (CLM4.5; Oleson et al.,
2013), for example, fluxes of heat and moisture occur from the leaves to the canopy air, from the
ground to the canopy air, and from the canopy air to the atmosphere (Figure 1a). The flux from
the canopy to the atmosphere is parameterized using Monin–Obukhov similarity theory (MOST).
This theory requires the displacement height ($d$) and roughness length ($z_0$). A challenge has
been to specify these, which are complex functions of the flow and physical canopy structure



(Shaw and Pereira 1982); simple parameterizations calculate them as a fixed fraction of canopy
height (as in the CLM4.5) or use relationships with leaf area index (Sellers et al., 1986;
Choudhury and Monteith, 1988; Raupach, 1994). An additional challenge, largely ignored in
land surface models, is that MOST fails in the roughness sublayer (RSL) extending to twice the
canopy height or more (Garratt, 1978; Physick and Garratt, 1995; Harman and Finnigan, 2007,
2008). While MOST successfully relates mean gradients and turbulent fluxes in the surface layer
above the RSL, within the RSL vertical fluxes are larger than expected from mean gradients
obtained using MOST.

Dual-source land surface models also require parameterization of turbulent processes

within the canopy, where wind speed regulates vegetation fluxes through the leaf boundary layer
conductance and where turbulent transport regulates fluxes between the ground and canopy air.
Following BATS (Dickinson et al., 1986), the CLM4.5 uses an ad-hoc parameterization without
resolving within-canopy profiles of wind speed or turbulence. Wind speed within the canopy is
taken as equal to the friction velocity ($u_*$), and the aerodynamic conductance between the ground
and canopy air is proportional to $u_*$. Zeng et al. (2005) subsequently modified this expression to
account for sparse and dense canopies.

Harman and Finnigan (2007, 2008) proposed a formulation by which traditional MOST

can be modified to account for the RSL. Their theoretical derivations couple the above-canopy
turbulent fluxes with equations for the mass and momentum balances within the canopy. Here,
we implement and test the theory in a multi-layer canopy model (Bonan et al., 2014). The
development of a multi-layer canopy for the ORCHIDEE land surface model has renewed
interest in the practical use of this class of canopy models (Ryder et al., 2016; Chen et al., 2016).
The earlier multi-layer model development of Bonan et al. (2014) focused on linking stomatal





conductance and plant hydraulics and neglected turbulent processes in the canopy. The current
work extends the model to include canopy-induced turbulence. The RSL theory avoids a priori
specification of $z_0$ and $d$ by linking these to canopy density and characteristics of the flow;
provides consistent forms for various turbulent terms above and within the canopy (friction
velocity, wind speed, scalar transfer coefficients); and provides a method for determining the
associated profiles of canopy air temperature and water vapor concentration. This study is
motivated by the premise that land surface models generally neglect canopy-induced turbulence,
that inclusion of this is critical to model simulations, and that the Harman and Finnigan (2007,
2008) RSL theory provides a tractable parameterization extending from the ground through the
canopy and the RSL.

**2 Methods**
We evaluated the canopy model at 12 AmeriFlux sites comprising 81 site-years of data using the
same protocol of the earlier model development (Bonan et al., 2014). We used the 6 forests sites
previously described in Bonan et al. (2014) and included additional flux data for 1 forest (US-
Dk2), 2 grassland (US-Dk1, US-Var), and 3 cropland sites (US-ARM, US-Bo1, US-Ne3) to test
the canopy model over a range of tall and short canopies, dense and sparse leaf area index, and
different climates (Table 1). Tower forcing data were from the North American Carbon Program
(NACP) site synthesis (Schaefer et al., 2012) as described previously (Bonan et al., 2014), except
as noted below for the three Duke tower sites. The model was evaluated using tower
observations of net radiation, sensible heat flux, latent heat flux, and friction velocity obtained
from the AmeriFlux Level 2 data set (ameriflux.lbl.gov) and with gross primary production from
the NACP site synthesis (Schaefer et al., 2012). We limited the simulations to one particular



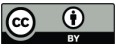

month (with the greatest leaf area) as in Bonan et al. (2014) so as to constrain the model without
having to account for seasonal changes in soil water.

Ryu et al. (2008) describe the US-Var grassland located in California. The CLM has been

previously tested using flux data from the US-Ne3 and US-Bo1 cropland sites (Levis et al.,
2012), and we used the same sites here. The US-Ne3 tower site is a rainfed maize (*Zea mays*) –
soybean (*Glycine max*) rotation located in Nebraska (Verma et al., 2005). We used flux data for
soybean, a $C_3$ crop (years 2002 and 2004). Kucharik and Twine (2007) give leaf area index, also
in the AmeriFlux biological, ancillary, disturbance and metadata. The same ancillary data show a
canopy height of 0.9 m during August for soybean. The US-Bo1 site is a maize–soybean rotation
located in Illinois (Meyers and Hollinger, 2004; Hollinger et al., 2005). Meyers and Hollinger
(2004) give canopy data. We used a leaf area index of 5 $m^2$ $m^{-2}$ and canopy height of 0.9 m for
soybean (1998–2006, even years). Flux data for the US-ARM winter wheat site, used to test the
CLM4.5, provides an additional dataset with which to test the model (Lu et al., 2017).

Stoy et al. (2006) provide site information for the US-Dk2 deciduous broadleaf forest

tower site located in the Duke Forest, North Carolina, which was included here to contrast the
adjacent evergreen needleleaf forest and grassland sites. The US-Dk1 tower site in the Duke
Forest provides an additional test for grassland (Novick et al., 2004; Stoy et al., 2006). Tower
forcing and flux data for 2004–2008 were obtained directly from the tower site investigators
(Kim Novick, personal communication).

**2.1 Model formulation**
The canopy model has three main components: leaf gas exchange and plant hydraulics; a
numerical solution for scalar profiles within and above the canopy; and inclusion of the RSL



parameterization. It builds upon the work of Bonan et al. (2014), which describes leaf gas
exchange and plant hydraulics for a multi-layer canopy with sunlit and shaded leaves at each
layer in the canopy. Radiative transfer of visible, near-infrared, and longwave radiation is
calculated at each level and accounts for forward and backward scattering within the canopy.
Bonan et al. (2014) used the radiative transfer model of Norman (1979). We retain that
parameterization for longwave radiation, but radiative transfer in the visible and near-infrared
wavebands is calculated from the two-stream approximation with the absorbed solar radiation
partitioned into direct beam, scattered direct beam, and diffuse radiation for sunlit and shaded
leaves in relation to cumulative plant area index as in Dai et al. (2004). This allows better
comparison with the CLM4.5, which uses the canopy-integrated two-stream solution for sunlit
and shaded leaves. The calculation of leaf temperature and fluxes is solved simultaneously with
stomatal conductance, photosynthesis, and leaf water potential in an iterative calculation. This
method numerically optimizes water-use efficiency within the constraints imposed by plant
water uptake to prevent leaf desiccation using the methodology of Williams et al. (1996). Soil
fluxes are calculated using the layer of canopy air immediately above the ground. Bonan et al.
(2014) provide further details.

Here, we describe the formulation of the scalar profiles and the RSL, which were not

included in Bonan et al. (2014). Figure 1 shows the numerical grid. The approach is conceptually
similar to the implementation of a multi-layer canopy in ORCHIDEE-CAN and that model's
implicit numerical coupling of leaf fluxes and scalar profiles (Ryder et al., 2016; Chen et al.,
2016), but modified to include sunlit and shaded leaves at each layer in the canopy and also the
RSL (Harman and Finnigan 2007, 2008). The grid spacing ($\Delta z$) is 0.5 m for forest and 0.1 m for
crop and grassland. We use thin layers to represent the light gradients that drive variation in leaf





water potential in the canopy as in Bonan et al. (2014). Indeed, it is this strong variation in leaf
water potential from the top of the canopy to the bottom that motivates the need for a multi-layer
canopy. Appendix A provides a complete description of the model, and Appendix B lists all
model variables.

**2.1.1 The coupled flux–profile equations**
In the volume of air extending from the ground to some reference height above the canopy, the
scalar conservation equations for heat and water vapor, the energy balances of the sunlit and
shaded canopy, and the ground energy balance provide a system of equations that can be solved
for air temperature, water vapor concentration, sunlit and shaded leaf temperatures, and ground
temperature. The scalar conservation equation for heat relates the change over some time interval
of air temperature ($\theta$, K) at height $z$ (m) to the source fluxes of sensible heat from the sunlit and
shaded portions of the canopy ($H_{\ell sun}$ and $H_{\ell sha}$, W m$^{-2}$) and the vertical flux ($H$, W m$^{-2}$). For a
vertically-resolved canopy, the one-dimensional conservation equation for temperature is
$$\rho_m c_p \frac{\partial \theta(z)}{\partial t} + \frac{\partial H}{\partial z} = \left[ H_{\ell sun}(z) f_{sun}(z) + H_{\ell sha}(z) \{1 - f_{sun}(z)\} \right] a(z) \qquad (1)$$
The equivalent equation for water vapor ($q$, mol mol$^{-1}$) in relation to the canopy source fluxes
($E_{\ell sun}$ and $E_{\ell sha}$, mol H$_2$O m$^{-2}$ s$^{-1}$) and vertical flux ($E$, mol H$_2$O m$^{-2}$ s$^{-1}$) is
$$\rho_m \frac{\partial q(z)}{\partial t} + \frac{\partial E}{\partial z} = \left[ E_{\ell sun}(z) f_{sun}(z) + E_{\ell sha}(z) \{1 - f_{sun}(z)\} \right] a(z) \qquad (2)$$
In this notation, $\rho_m$ is molar density (mol m$^{-3}$) and $c_p$ is the specific heat of air (J mol$^{-1}$ K$^{-1}$).
$a(z)$ is the plant area density, which is equal to the leaf and stem area increment of a canopy
layer divided by the thickness of the layer ($\Delta L(z) / \Delta z$ ; m$^2$ m$^{-3}$), and $f_{sun}$ is the sunlit fraction of



the layer.  As in Harman and Finnigan (2007, 2008), the vertical fluxes are parameterized using a
first-order turbulence closure (K-theory) whereby the sensible heat flux is
$$H(z) = -\rho_m c_p K_c(z) \frac{\partial \theta}{\partial z} \tag{3}$$

and the water vapor flux is
$$E(z) = -\rho_m K_c(z) \frac{\partial q}{\partial z} \tag{4}$$

with $K_c$ the scalar diffusivity ($m^2\ s^{-1}$), assumed to be the same for heat and water vapor. These
equations apply above and within the canopy, but with $a(z) = 0$ for layers without vegetation.

The source fluxes of sensible heat and water vapor are described by the energy balance

equation and are provided separately for sunlit and shaded fractions of the canopy layer. The
energy balance of sunlit leaves at height $z$ in the canopy is
$$c_L(z) \frac{\partial T_{\ell sun}(z)}{\partial t} \Delta L_{sun}(z) = \left[ R_{n\ell sun}(z) - H_{\ell sun}(z) - \lambda E_{\ell sun}(z) \right] \Delta L_{sun}(z) \tag{5}$$

The left-hand side is the storage of heat ($W\ m^{-2}$) in a layer of vegetation with heat capacity $c_L$ (J
$m^{-2}\ K^{-1}$), temperature $T_{\ell sun}$ (K), and plant area index $\Delta L_{sun} = f_{sun} \Delta L$ ($m^2\ m^{-2}$). The right-hand
side is the balance between net radiation ($R_{n\ell sun}$; positive denotes energy gain), sensible heat flux
($H_{\ell sun}$; positive away from the leaf), and latent heat flux ($\lambda E_{\ell sun}$; positive away from the leaf).
The sensible heat flux is
$$H_{\ell sun}(z) = 2c_p \left[ T_{\ell sun}(z) - \theta(z) \right] g_b(z) \tag{6}$$

and the evapotranspiration flux is
$$E_{\ell sun}(z) = \left[ q_{sat}(T_{\ell sun}) - q(z) \right] g_{\ell sun}(z) \tag{7}$$





For sensible heat, $g_b$ is the leaf boundary layer conductance (mol m$^{-2}$ s$^{-1}$), and the factor two
appears because heat transfer occurs from both sides of plant material. The evapotranspiration
flux depends on the saturated water vapor concentration of the leaf, which varies with leaf
temperature and is denoted as $q_{sat}(T_{\ell sun})$. It also requires a leaf conductance ($g_{\ell sun}$, mol m$^{-2}$ s$^{-1}$)
that combines evaporation from the wetted fraction of the canopy and transpiration from the dry
fraction. A similar equation applies to shaded leaves. The energy balance given by Eq. (5) does
not account for snow in the canopy, so the simulations are restricted to snow-free periods.

These equations are discretized in space and time and are solved in an implicit system of

equations for time $n+1$. Ryder et al. (2016) and Chen et al. (2016) describe the solution using a
single leaf. Here, the solution is given for separate sunlit and shaded portions of the canopy. In
numerical form and with reference to Figure 1, the scalar conservation equation for temperature
is
$$
\begin{aligned}
&\frac{\rho_m \Delta z_i}{\Delta t} c_p \left( \theta_i^{n+1} - \theta_i^n \right) - g_{a,i-1} c_p \theta_{i-1}^{n+1} + \left( g_{a,i-1} + g_{a,i} \right) c_p \theta_i^{n+1} - g_{a,i} c_p \theta_{i+1}^{n+1} = \\
&2 g_{b,i} c_p \left( T_{\ell sun,i}^{n+1} - \theta_i^{n+1} \right) \Delta L_{sun,i} + 2 g_{b,i} c_p \left( T_{\ell sha,i}^{n+1} - \theta_i^{n+1} \right) \Delta L_{sha,i}
\end{aligned}
\tag{8}
$$

and for water vapor is
$$
\begin{aligned}
&\frac{\rho_m \Delta z_i}{\Delta t} \left( q_i^{n+1} - q_i^n \right) - g_{a,i-1} q_{i-1}^{n+1} + \left( g_{a,i-1} + g_{a,i} \right) q_i^{n+1} - g_{a,i} q_{i+1}^{n+1} = \\
&\left[ q_{sat}\left( T_{\ell sun,i}^n \right) + s_i^{sun} \left( T_{\ell sun,i}^{n+1} - T_{\ell sun,i}^n \right) - q_i^{n+1} \right] g_{\ell sun,i} \Delta L_{sun,i} + \\
&\left[ q_{sat}\left( T_{\ell sha,i}^n \right) + s_i^{sha} \left( T_{\ell sha,i}^{n+1} - T_{\ell sha,i}^n \right) - q_i^{n+1} \right] g_{\ell sha,i} \Delta L_{sha,i}
\end{aligned}
\tag{9}
$$

The first term on the left-hand side of Eq. (8) is the storage of heat (W m$^{-2}$) over the time interval
$\Delta t$ (s) in a layer of air with thickness $\Delta z_i$ (m). The next three terms describe the vertical fluxes
from Eq. (3). These use conductance notation in which $g_a$ is an aerodynamic conductance (mol
m$^{-2}$ s$^{-1}$) that is nominally related to $\rho_m K_c / \Delta z$ (Eq. (25) provides the formal relationship). $g_{a,i}$ is

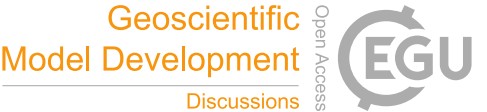



the aerodynamic conductance between layer $i$ to $i+1$ above, and $g_{a,i-1}$ is the similar
conductance below between layer $i$ to $i-1$. The two terms on the right-hand side of Eq. (8) are
the vegetation source fluxes of sensible heat for the sunlit and shaded portions of the canopy
layer. Eq. (9) uses comparable terms for water vapor, with $q_{sat}(T_{\ell sun})$ and $q_{sat}(T_{\ell sha})$ linearized as
explained below.
The sunlit and shaded temperatures required for Eqs. (8) and (9) are obtained from the
energy balance at canopy layer $i$. For the sunlit portion of the canopy
$$\frac{c_{L,i}}{\Delta t}\left(T_{\ell sun,i}^{n+1} - T_{\ell sun,i}^{n}\right) = R_{n\ell sun,i} - 2g_{b,i}c_{p}\left(T_{\ell sun,i}^{n+1} - \theta_{i}^{n+1}\right)$$
$$-\lambda\left[q_{sat}\left(T_{\ell sun,i}^{n}\right) + s_{i}^{sun}\left(T_{\ell sun,i}^{n+1} - T_{\ell sun,i}^{n}\right) - q_{i}^{n+1}\right]g_{\ell sun,i}$$
(10)

Latent heat flux uses the linear approximation
$$q_{sat}\left(T_{\ell sun,i}^{n+1}\right) = q_{sat}\left(T_{\ell sun,i}^{n}\right) + s_{i}^{sun}\left(T_{\ell sun,i}^{n+1} - T_{\ell sun,i}^{n}\right)$$
(11)

with $s_{i}^{sun} = dq_{sat}/dT$ evaluated at $T_{\ell sun,i}^{n}$. The leaf boundary layer conductance ($g_{b,i}$) depends on
wind speed ($u_{i}$, m s$^{-1}$) as described by Bonan et al. (2014). The conductance for transpiration is
equal to the leaf boundary layer and stomatal conductances acting in series, i.e., $(g_{b,i}^{-1} + g_{sun,i}^{-1})^{-1}$.
Here, it is assumed that $g_{b,i}$ is the same for heat and water vapor (as in the CLM4.5). Stomatal
conductance ($g_{sun,i}$) is calculated based on water-use efficiency optimization and plant
hydraulics (Bonan et al., 2014). The total conductance ($g_{\ell sun,i}$) combines evaporation from the
wetted fraction of the plant material ($f_{wet,i}$) and transpiration from the dry fraction ($f_{dry,i}$),
similar to that in the CLM4.5 in which
$$g_{\ell sun,i} = \left(\frac{g_{sun,i}g_{b,i}}{g_{sun,i} + g_{b,i}}\right)f_{dry,i} + g_{b,i}f_{wet,i}$$
(12)





with $f_{dry,i} = f_{green,i}(1 - f_{wet,i})$ so that interception occurs from stems and leaves, but transpiration
occurs only from green leaves (denoted by the green leaf fraction $f_{green,i}$). The comparable
equation for shaded leaves is
$$\frac{c_{L,i}}{\Delta t}\left(T_{\ell sha,i}^{n+1} - T_{\ell sha,i}^{n}\right) = R_{n\ell sha,i} - 2c_p\left(T_{\ell sha,i}^{n+1} - \theta_i^{n+1}\right)g_{b,i}$$
$$-\lambda\left[q_{sat}\left(T_{\ell sha,i}^{n}\right) + s_i^{sha}\left(T_{\ell sha,i}^{n+1} - T_{\ell sha,i}^{n}\right) - q_i^{n+1}\right]g_{\ell sha,i}$$
(13)

We use post-CLM4.5 changes in intercepted water ($W$, kg m$^{-2}$) and the wet and dry fractions of
the canopy ($f_{wet}$, $f_{dry}$) that are included in the next version of the model (CLM5).
At the lowest layer above the ground ($i = 1$), the ground fluxes $H_0$ and $E_0$ are additional
source fluxes, and the ground surface energy balance must be solved to provide the ground
temperature ($T_0^{n+1}$, K). This energy balance is
$$R_{n0} = c_p\left(T_0^{n+1} - \theta_1^{n+1}\right)g_{a,0} + \lambda\left\{h_{s0}\left[q_{sat}\left(T_0^{n}\right) + s_0\left(T_0^{n+1} - T_0^{n}\right)\right] - q_1^{n+1}\right\}g_{s0}$$
$$+ \frac{\kappa_{soil}}{\Delta z_{soil}}\left(T_0^{n+1} - T_{soil}^{n}\right)$$
(14)

The first term on the right-hand side is the sensible heat flux between the ground with
temperature $T_0$ and the air in the canopy layer immediately above the ground with temperature
$\theta_1$; $g_{a,0}$ is the corresponding aerodynamic conductance. The second term is the latent heat flux,
with $q_1$ the water vapor concentration of the canopy air. In calculating soil evaporation, the
surface water vapor concentration is
$q_0^{n+1} = h_{s0}q_{sat}\left(T_0^{n+1}\right) = h_{s0}\left[q_{sat}\left(T_0^{n}\right) + s_0\left(T_0^{n+1} - T_0^{n}\right)\right]$  (15)
with $s_0 = dq_{sat}/dT$ evaluated at $T_0^{n}$. Evaporation depends on the fractional humidity of the first
soil layer ($h_{s0}$; CLM5). The soil evaporative conductance ($g_{s0}$) is the total conductance and



consists of the aerodynamic conductance ($g_{a,0}$) and a soil surface conductance to evaporation
($g_{soil}$; CLM5) acting in series. The last term in Eq. (14) is the heat flux to the soil, which
depends on the thermal conductivity ($\kappa_{soil}$), thickness ($\Delta z_{soil}$), and temperature ($T_{soil}$) of the
first soil layer. Eq. (14) does not account for snow on the ground, and the simulations are
restricted to snow-free periods.

The numerical solution involves rewriting Eqs. (10) and (13) to obtain expressions for

$T_{\ell sun,i}^{n+1}$ and $T_{\ell sha,i}^{n+1}$ and substituting these in Eqs. (8) and (9). Eqs. (14) and (15) provide the
necessary expressions for $T_0^{n+1}$ and $q_0^{n+1}$ at $i=1$. This gives a tridiagonal system of implicit
equations with the form
$$a_{1,i}\theta_{i-1}^{n+1} + b_{11,i}\theta_i^{n+1} + b_{12,i}q_i^{n+1} + c_{1,i}\theta_{i+1}^{n+1} = d_{1,i} \tag{16}$$
$$a_{2,i}q_{i-1}^{n+1} + b_{21,i}\theta_i^{n+1} + b_{22,i}q_i^{n+1} + c_{2,i}q_{i+1}^{n+1} = d_{2,i} \tag{17}$$
in which $a_{1,i}$, $a_{2,i}$, $b_{11,i}$, $b_{21,i}$, $b_{12,i}$, $b_{22,i}$, $c_{1,i}$, $c_{2,i}$, $d_{1,i}$, and $d_{2,i}$ are algebraic coefficients
(Appendix A1). The system of equations is solved using the method of Richtmyer and Morton
(1967, pp. 275–278), as described in Sect. S1 of the Supplement. $\theta_i^{n+1}$ and $q_i^{n+1}$ are obtained for
each level with the boundary conditions $\theta_{ref}^{n+1}$ and $q_{ref}^{n+1}$ the temperature and water vapor
concentration at some reference height above the canopy. Then, the leaf temperatures and fluxes
and ground temperature and fluxes are evaluated. Ryder et al. (2016) used a different, but
algebraically equivalent, solution in their model.

The equation set has several dependencies that preclude a fully implicit solution for $\theta_i^{n+1}$,

$q_i^{n+1}$, $T_{\ell sun,i}^{n+1}$, $T_{\ell sha,i}^{n+1}$, and $T_0^{n+1}$. Net radiation depends on leaf and ground temperatures. Ryder et al.
(2016) avoided this by specifying longwave emission as an implicit term in the source energy





balance equation, but there are other complicating factors. Boundary layer conductance is
calculated from wind speed, but also air and leaf temperatures (to account for free convection
using the Grashof number). The wet and dry fractions of the canopy vary with evaporative flux.
Wind speed and aerodynamic conductances depend on the surface layer stability as quantified by
the Obukhov length, yet this length scale depends on the surface fluxes. Stomatal conductance
requires leaf temperature, air temperature, and water vapor concentration. Further complexity to
the canopy flux calculations arises because stomatal conductance is calculated from principles of
water transport along the soil–plant–atmosphere continuum such that leaf water potential cannot
drop below some threshold (Williams et al., 1996; Bonan et al., 2014). This requires the leaf
transpiration flux, which itself depends on stomatal conductance. The CLM4.5 has similar
dependences in its surface flux calculation and solves the fluxes in a numerical procedure with
up to 40 iterations for a single model timestep. Instead, we solve the equations using a 5-minute
sub-timestep to evaluate fluxes over a full model timestep (30 minutes when coupled to an
atmospheric model). In the sub-timestep looping, the current values of wind speed, temperature,
water vapor concentration, and canopy water are used to calculate the leaf and aerodynamic
conductances needed to update the flux–profiles.

**2.1.2 Plant canopy and roughness sublayer**
The solution to the scalar fluxes and profiles described in the preceding section requires the
aerodynamic conductance ($g_a$), and also wind speed ($u$) to calculate leaf boundary layer
conductance ($g_b$). These are provided by the RSL parameterization. We follow the theory of
Harman and Finnigan (2007, 2008). In their notation, the coordinate system is defined such that
the vertical origin is the top of the canopy and $z$ is the deviation from the canopy top. Here, we

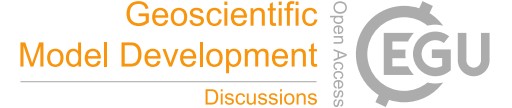



retain $z$ as the physical height above the ground, whereby $z - h$ is the deviation from the
canopy top. The Harman and Finnigan (2007, 2008) parameterization modifies the MOST
profiles of $u$, $\theta$, and $q$ above plant canopies for the RSL and does not require a multi-layer
canopy (e.g., Harman, 2012), but was derived by coupling the above-canopy momentum and
scalar fluxes with equations for the momentum and scalar balances within a dense, horizontally
homogenous canopy. Here, we additionally utilize the within-canopy equations.

Neglecting the RSL, the wind speed profile is described by MOST as

$$u(z) = \frac{u_*}{k} \left[ \ln\left(\frac{z-d}{z_0}\right) - \psi_m\left(\frac{z-d}{L_{MO}}\right) + \psi_m\left(\frac{z_0}{L_{MO}}\right) \right] \qquad (18)$$
where $u_*$ is friction velocity (m s$^{-1}$), $z$ is height above the ground (m), $d$ is displacement height
(m), $z_0$ is roughness length (m), and the similarity function $\psi_m$ adjusts the log profile in relation
to the Obukhov length ($L_{MO}$, m). The Harman and Finnigan (2007, 2008) RSL parameterization
reformulates this as
$$u(z) = \frac{u_*}{k} \left[ \ln\left(\frac{z-d}{h-d}\right) - \psi_m\left(\frac{z-d}{L_{MO}}\right) + \psi_m\left(\frac{h-d}{L_{MO}}\right) + \hat{\psi}_m\left(\frac{z-d}{L_{MO}}, \frac{z-d}{l_m/\beta}\right) - \hat{\psi}_m\left(\frac{h-d}{L_{MO}}, \frac{h-d}{l_m/\beta}\right) + \frac{k}{\beta} \right] \quad (19)$$
This equation is analogous to the previous equation, but is valid only for wind speed above the
canopy at heights $z \geq h$. It rewrites Eq. (18) so that the lower surface is the canopy height ($h$,
m) rather than the apparent sink for momentum ($d + z_0$). This eliminates $z_0$, but introduces $u(h)$
(the wind speed at the top of the canopy) as a new term, which is specified by $\beta = u_* / u(h)$. Eq.
(19) also introduces $\hat{\psi}_m$, which adjusts the profile to account for canopy-induced physics in the
RSL. Whereas $\psi_m$ uses the length scale $L_{MO}$, $\hat{\psi}_m$ introduces a second length scale $l_m / \beta$. This
length scale is the dominant scale of the shear-driven turbulence generated at or near the canopy





top, is equal to $u / (\partial u / \partial z)$ at the top of the canopy, and relates to canopy density. The
corresponding equation for temperature above the canopy is
$\theta(z) - \theta(h) = \dfrac{\theta_*}{k}\left[\ln\left(\dfrac{z-d}{h-d}\right) - \psi_c\left(\dfrac{z-d}{L_{MO}}\right) + \psi_c\left(\dfrac{h-d}{L_{MO}}\right) + \hat{\psi}_c\left(\dfrac{z-d}{L_{MO}}, \dfrac{z-d}{l_m/\beta}\right) - \hat{\psi}_c\left(\dfrac{h-d}{L_{MO}}, \dfrac{h-d}{l_m/\beta}\right)\right]$ (20)
with $\theta_*$ a temperature scale (K) and $\psi_c$ and $\hat{\psi}_c$ corresponding functions for scalars. The same
equation applies to water vapor, but substituting $q$ and $q_*$. The new terms in the profile
equations introduced by the RSL theory are: $\beta$, the ratio of friction velocity to wind speed at the
canopy height; $l_m$, the mixing length (m) in the canopy; and the modified similarity functions
$\hat{\psi}_m$ and $\hat{\psi}_c$. Expressions for these are obtained by considering the momentum and scalar
balances within a dense, horizontally homogenous canopy and by matching the above- and
within-canopy profile equations at the canopy height $h$ (Appendix A2). In addition, the RSL
theory provides an equation for $d$, rather than specifying this as an input parameter. Eq. (20)
also requires $\theta(h)$, the air temperature (K) at the canopy height. Harman and Finnigan (2008)
provide an equation that relates this to the bulk surface temperature ($\theta_s$) for use with a bulk
surface parameterization. Here, we treat $\theta(h)$ as a prognostic variable obtained for the top
canopy layer as described in the previous section.
With the assumption of a constant mixing length ($l_m$) in the canopy, wind speed within
the canopy at heights $z \le h$ follows an exponential decline with greater depth in the canopy in
relation to the height $z - h$ normalized by the length scale $l_m / \beta$, with
$u(z) = u(h)\exp\left[\dfrac{z-h}{l_m/\beta}\right]$ (21)





This is the same equation derived by Inoue (1963) and Cionco (1965), but they express the
exponential term as $-\eta(1 - z / h)$, where $\eta$ is an empirical parameter. Harman and Finnigan
(2007, 2008) introduced the notation $l_m / \beta$, whereby $\eta / h = \beta / l_m$, so that the exponential decay
of wind speed in the canopy relates to the RSL. The wind speed profile matches Eq. (19) at the
top of the canopy through $u(h)$. We restrict $u \geq 0.1$ m s$^{-1}$ (see Discussion for further details).
The corresponding profile for the scalar diffusivity within the canopy is similar to that for wind
with
$$K_c\left(z\right) = K_c\left(h\right)\exp\left[\frac{z - h}{l_m / \beta}\right] \tag{22}$$
In the RSL theory of Harman and Finnigan (2008),
$$K_c(h) = l_m\, u_* / S_c \tag{23}$$
where the Schmidt number ($S_c$) is defined as the ratio of the diffusivities for momentum and
scalars at the top of the canopy (Appendix A2). The diffusivity of water vapor is assumed to
equal that for heat as in Harman and Finnigan (2008). Eq. (21) for $u$ and Eq. (22) for $K_c$ are
derived from first-order turbulence closure with constant mixing length in the canopy. They have
been used previously to parameterize within-canopy wind and scalar diffusivity in plant canopy
models (Shuttleworth and Wallace, 1985; Choudhury and Monteith, 1988), land surface models
(Dolman, 1993; Bonan, 1996; Niu and Yang, 2004), and hydrologic models (Mahat et al., 2013;
Clark et al., 2015), but without the RSL and with $\eta$ specified as a model parameter.
The aerodynamic conductance for scalars at level $i$ above the canopy ($z > h$) between
heights $z_i$ and $z_{i+1}$ is
$$g_{a,i} = \rho_m k u_* \left[\ln\left(\frac{z_{i+1} - d}{z_i - d}\right) - \psi_c\left(\frac{z_{i+1} - d}{L_{MO}}\right) + \psi_c\left(\frac{z_i - d}{L_{MO}}\right) + \hat{\psi}_c\left(z_{i+1}\right) - \hat{\psi}_c\left(z_i\right)\right]^{-1} \tag{24}$$





where $\hat{\psi}_c$ is evaluated at $z_i$ and $z_{i+1}$. The conductance within the canopy ($z < h$) consistent with
the RSL theory is obtained from Eq. (22) as
$$\frac{1}{g_{a,i}} = \frac{1}{\rho_m} \int_{z_i}^{z_{i+1}} \frac{dz}{K_c(z)} \tag{25}$$

so that
$$\frac{1}{g_{a,i}} = \frac{1}{\rho_m} \frac{S_c}{\beta u_*} \left\{ \exp\left[ -\frac{(z_i - h)}{l_m / \beta} \right] - \exp\left[ -\frac{(z_{i+1} - h)}{l_m / \beta} \right] \right\} \tag{26}$$

For the top canopy layer, the conductance is integrated between the heights $z_i$ and $h$, and the
above-canopy conductance from $h$ to $z_{i+1}$ is additionally included. The conductance
immediately above the ground is
$$g_{a,0} = \rho_m k^2 u_1 \left[ \ln\left( \frac{z_1}{z_{0m}} \right) \ln\left( \frac{z_1}{z_{0c}} \right) \right]^{-1} \tag{27}$$

with $z_{0m} = 0.01$ m and $z_{0c} = 0.1 z_{0m}$ the roughness lengths of the ground for momentum and
scalars, respectively, and assuming neutral stability in this layer. In calculating the conductances,
we use the constraint $\rho_m / g_{a,i} \leq 500$ s m$^{-1}$ (see Discussion for further details).

Harman and Finnigan (2007, 2008) provide a complete description of the RSL equations

and their derivation. Appendix A2 gives the necessary equations as implemented herein. Use of
the RSL parameterization requires specification of the Monin–Obukhov functions $\psi_m$ and $\psi_c$,
the RSL functions $\hat{\psi}_m$ and $\hat{\psi}_c$, and equations for $\beta$ and $S_c$. Expressions for $l_m$ and $d$ are
obtained from $\beta$. Solution to the RSL parameterization requires an iterative calculation for the
Obukhov length ($L_{MO}$) as shown in Figure 2 and explained further in Appendix A3. The



equations as described above apply to dense canopies. Appendix A4 gives a modification for
sparse canopies.

**2.1.3 Plant area density**
Land surface models commonly combine leaf and stem area into a single plant area index to
calculate radiative transfer, and the CLM4.5 does the same. By using plant area index, big-leaf
canopy models assume that woody phytoelements (branches, stems) are randomly interspersed
among leaves. Some studies of forest canopies suggest that branches and stems are shaded by
foliage and therefore contribute much less to obscuring the sky than if they were randomly
dispersed among foliage (Norman and Jarvis, 1974; Kucharik et al., 1998). To allow for shading,
we represent plant area density as separate profiles of leaf and stem area. The beta distribution
probability density function provides a continuous profile of leaf area density for use with multi-
layer canopy models, and we use a uniform profile for stem area, whereby
$$a(z) = \frac{L_T}{h} \frac{(z/h)^{p-1}(1-z/h)^{q-1}}{\mathrm{B}(p,q)} + \frac{S_T}{h}$$    (28)
The first term on the right-hand side is the leaf area density with $z/h$ the relative height in the
canopy and $L_T$ leaf area index (m$^2$ m$^{-2}$). The beta function ($\mathrm{B}$) is a normalization constant. The
parameters $p$ and $q$ determine the shape of the profile (Figure 3). Representative values are
$p = q = 2.5$ for grassland and cropland, $p = 3.5$ and $q = 2.0$ for deciduous trees and spruce
trees, and $p = 11.5$ and $q = 3.5$ for pine trees (Meyers et al., 1998; Wu et al., 2003). The second
term on the right-hand side is the stem area density calculated from the stem area index of the
canopy ($S_T$). For these simulations, $L_T$ comes from tower data (Table 1), and $S_T$ is estimated
from $L_T$ as in the CLM4.5.






### 2.1.4 Leaf heat capacity

The CLM4.5 requires specific leaf area as an input parameter, and we use this to calculate leaf
heat capacity (per unit leaf area). Specific leaf area, as used in the CLM4.5, is the area of a leaf
per unit mass of carbon ($m^2$ $g^{-1}$ C) and is the inverse of leaf carbon mass per unit area ($M_a$, g C
$m^{-2}$). This latter parameter is converted to dry mass assuming the carbon content of dry biomass
is 50% so that the leaf dry mass per unit area is $M_a / f_c$ with $f_c = 0.5$ g C $g^{-1}$. The leaf heat
capacity ($c_L$, J $m^{-2}$ $K^{-1}$) is calculated from leaf dry mass per unit area after adjusting for the mass
of water, as in Ball et al. (1988) and Blanken et al. (1997). Following Ball et al. (1988), we
assume that the specific heat of dry biomass is one-third that of water ($c_{dry} = 1.396$ J $g^{-1}$ $K^{-1}$).
Then, with $f_w$ the fraction of fresh biomass that is water, the leaf heat capacity is
$$c_L = \frac{M_a}{f_c} c_{dry} + \frac{M_a}{f_c}\left(\frac{f_w}{1-f_w}\right) c_{wat} \tag{29}$$
The first term on the right-hand side is the mass of dry biomass multiplied by the specific heat of
dry biomass. The second term is the mass of water multiplied by the specific heat of water
($c_{wat} = 4.188$ J $g^{-1}$ $K^{-1}$). We assume that 70% of fresh biomass is water ($f_w = 0.7$ g $H_2O$ $g^{-1}$).
Niinemets (1999) reported a value of 0.66 g $H_2O$ $g^{-1}$ in an analysis of leaves from woody plants.
The calculated heat capacity for grasses, crops, and trees is 745–2792 J $m^{-2}$ $K^{-1}$ depending on
specific leaf area (Table 2). For comparison, Blanken et al. (1997) calculated a heat capacity of
1999 J $m^{-2}$ $K^{-1}$ for aspen leaves with a leaf mass per area of 111 g $m^{-2}$ and $f_w = 0.8$. Ball et al.
(1988) reported a range of 1100–2200 J $m^{-2}$ $K^{-1}$ for mangrove leaves spanning a leaf mass per
area of 93–189 g $m^{-2}$ with $f_w = 0.71$.






**2.2 Model simulations**

We performed several model simulations to compare the CLM4.5 with the RSL enabled multi-layer canopy and to incrementally evaluate the effect of specific processes on model performance. Table 3 summarizes the major model differences, and Table 4 summarizes the model simulations. The simulations discussed herein are:

1. CLM4.5 – Simulations with the CLM4.5 using tower meteorology and site data for leaf area index, stem area index, and canopy height.

2. m0 – This uses the multi-layer canopy, but configured to be similar to the CLM4.5 for leaf biophysics as described in Table 3. Stomatal conductance is calculated as in the CLM4.5. Leaf nitrogen declines exponentially with greater cumulative plant area index from the canopy top with the decay coefficient $K_n = 0.3$ as in the CLM4.5. The nitrogen profile determines the photosynthetic capacity at each layer so that leaves in the upper canopy have greater maximum photosynthetic rates than leaves in the lower canopy. In addition, leaf and stem area are comingled in the CLM4.5, and there is no heat storage in plant biomass. These features are replicated by having a uniform plant area density profile and by setting leaf heat capacity to a small, non-zero number. This simulation excludes a turbulence parameterization so that air temperature, water vapor concentration, and wind speed in the canopy are equal to the reference height forcing. Juang et al. (2008) referred to this as the well-mixed assumption. In this configuration, the fluxes of sensible and latent heat above the canopy are the sum of the source fluxes in the canopy, and friction velocity is not calculated. This is the baseline model configuration.

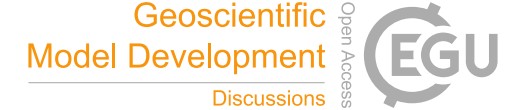



3. m1 – As in m0, but introducing a turbulence closure in the absence of the RSL. Eqs. (16) and
(17) are used to calculate $\theta$ and $q$. The CLM4.5 MOST parameterization is used to
calculate $u$ and $g_a$ above the canopy. Within the canopy, the mixing length model with
exponential profiles for $u$ and $g_a$ as in Eqs. (21) and (26) is used, but with $\eta = 3$, which is a
representative value found in many observational studies of wind speed in plant canopies
(Thom, 1975; Cionco, 1978; Brutsaert, 1982).
The multi-layer canopy model has several changes to leaf biophysics compared with the
CLM4.5. These differences are individually examined in the simulations:
4. b1 – As in m1, but with stomatal conductance calculated using water-use efficiency and plant
hydraulics as in Bonan et al. (2014).
5. b2 – As in b1, but with $K_n$ dependent on photosynthetic capacity ($V_{c\max}$) as in Bonan et al.

(2014).

6. b3 – As in b2, but with plant area density calculated from Eq. (28).
7. b4 – As in b3, but with leaf heat capacity from Eq. (29). This represents the full suite of
parameterization changes prior to inclusion of the RSL. We refer to this simulation also as
ML-RSL.
The final two simulations examine the RSL:
8. r1 – As in b4, but with the RSL parameterization used to calculate $u$ and $g_a$ above the
canopy using Eqs. (19) and (24). In this configuration, the CLM4.5 MOST parameterization
is replaced by the RSL parameterization for above-canopy profiles, but $\eta = 3$ for within
canopy profiles.





9.  r2 – As in r1, but $u$ and $g_a$ in the canopy are calculated from the RSL parameterization
using $l_m / \beta$ rather than $\eta = 3$. This is the full ML+RSL configuration, and comparison with
ML-RSL shows the effects of including the RSL parameterization.

Simulations were evaluated in terms of net radiation, sensible heat flux, latent heat flux,

gross primary production, friction velocity, and radiative temperature. Radiative temperature for
both the observations and simulations was evaluated from the upward longwave flux using an
emissivity of one. The simulations were assessed in terms of root mean square error (RMSE) for
each of the 81 site–years. We additionally assessed model performance using Taylor diagrams
and the corresponding skill score (Taylor, 2001) as in Bonan et al. (2014). Taylor diagrams
quantify the degree of similarity between the observed and simulated time series of a particular
variable in terms of the correlation coefficient ($r$) and the standard deviation of the model data
relative to that of the observations ($\hat{\sigma}$). The Taylor skill score combines these two measures into
a single metric of model performance with a value of one when $r = 1$ and $\hat{\sigma} = 1$.

**3 Results**
The ML+RSL simulation has better skill compared with CLM4.5 at most sites and for most
variables (Table 5). Of the 7 forest sites, net radiation ($R_n$) is improved at 5 sites, sensible heat
flux ($H$) at 5 sites, latent heat flux ($\lambda E$) at 4 sites, friction velocity ($u_*$) at 6 sites, radiative
temperature ($T_{rad}$) at the 5 sites with data, and gross primary production (GPP) at 3 of the 5 sites
with data. $H$ is improved at all 5 herbaceous sites, $\lambda E$ at 3 sites, $u_*$ at 3 sites, $T_{rad}$ at 4 sites,
and GPP at the 2 sites with data. $R_n$ generally is unchanged at the herbaceous sites.





Simulations for US-UMB illustrate these improvements for the forest sites, where the
influence of the RSL is greatest. For July 2006, CLM4.5 overestimates mid-day $H$ and
underestimates mid-day GPP (Figure 4). Mid-day latent heat flux is biased low, but within the
measurement error. $u_*$ is underestimated at night, and $T_{rad}$ has a larger diurnal range with colder
temperatures at night and warmer temperatures during the day compared with the observations.
ML+RSL improves the simulation. Mid-day $H$ decreases and GPP increases, nighttime $u_*$
increases, and the diurnal range of $T_{rad}$ decreases. Taylor diagrams for all years (1999–2006;
Figure 5) show improved $H$, $\lambda E$, and GPP (in terms of the variance of the modeled fluxes
relative to the observations), $u_*$ (in terms of correlation with the observations), and $T_{rad}$ (both
variance and correlation). Similar improvements are seen at the other forest sites.
The observations have a distinct relationship between $H$ and the temperature difference
between the surface and reference height ($T_{rad} - T_{ref}$), as shown in Figure 6 for two forest sites
(US-UMB and US-Me2) and one crop site (US-ARM) where the root mean square error of the
model (ML+RSL) is low for $H$ and $T_{rad}$. The observations show a positive correlation between
$T_{rad} - T_{ref}$ and $H$ beginning at about –2 °C. CLM4.5 and ML+RSL capture this relationship, but
the slope at the forest sites is smaller for CLM4.5 than for ML+RSL and the data have more
scatter. The observations show a complex relationship between temperature and $H$ for stable
conditions ($H < 0$). At the forest sites, CLM4.5 shows a slight linear increase in sensible heat
transfer to the surface (US-UMB) or is nearly invariant (US-Me2) as $T_{rad}$ becomes progressively
colder than $T_{ref}$. ML+RSL better captures the observations, particularly the more negative $H$ as
$T_{rad} - T_{ref}$ approaches zero. CLM4.5 also has a wider range of temperatures compared with the
observations and ML+RSL at the forest sites. Both models perform similarly at US-ARM.





Comparisons of ML-RSL and ML+RSL for US-UMB (July 2006) show improvements in
the multi-layer canopy even without the RSL parameterization (Figure 4). ML-RSL reduces mid-
day $H$, increases mid-day $\lambda E$ and GPP, and reduces the diurnal range of $T_{rad}$. The nighttime
bias in $u_*$ also decreases. Inclusion of the RSL (ML+RSL) further improves $u_*$ and $T_{rad}$, but
slightly degrades $H$ by increasing the daytime peak.
Comparison of the suite of simulations (m0 to r2; Table 4) for forest sites highlights the
effect of specific parameterization changes on model performance. The m0 simulation without a
turbulence closure has high RMSE compared with CLM4.5 for $\lambda E$ (Figure 7) and $H$ (Figure 8).
Inclusion of a turbulence closure (above-canopy, CLM4.5 MOST; within-canopy, mixing length
model) in m1 substantially reduces RMSE compared with m0 at all sites. The m1 RMSE for $\lambda E$
is reduced compared with CLM4.5 at 5 of the 7 sites and for $H$ at 4 sites. The leaf biophysical
simulations (b1–b4) reduce $\lambda E$ RMSE compared with m1 at 6 sites (US-Ho1 is the exception),
and the RMSE also decreases compared with CLM4.5 (Figure 7). Among b1–b4, the biggest
effect on $\lambda E$ RMSE occurs from stomatal conductance and nitrogen profiles (b1 and b2). The
RSL parameterization (r1 and r2) has relatively little additional effect on RMSE. The leaf
biophysical simulations (b1–b4) have a similar effect to reduce RMSE for $H$ compared with
m1, and RMSE decreases compared with CLM4.5 (Figure 8). Inclusion of the RSL (r1 and r2)
degrades $H$ in terms of RMSE. Whereas the b4 simulation without the RSL parameterization
decreases RMSE compared with CLM4.5, this reduction in RMSE is lessened in r1 and r2. The
RMSE for $u_*$ in m1 decreases compared with CLM4.5 at all sites (Figure 9). The leaf biophysics
simulations have little effect on RMSE, but the RSL simulations (r1 and r2) further reduce
RMSE.





The m0 simulation without a turbulence closure has substantially lower RMSE for $T_{rad}$

compared with the other simulations (Figure 10). This is seen in an improved simulation of the
diurnal temperature range, with warmer nighttime minimum and cooler daytime maximum
temperatures compared with the other simulations (not shown). The m1 simulation increases
RMSE, but RMSE is still reduced compared with CLM4.5 at the 5 sites with data. The leaf
biophysical simulations (b1–b4) have little effect on $T_{rad}$ , but the RSL simulations reduce
RMSE, more so for r1 than r2. Leaf temperature profiles are consistent with these results, as
shown in Figure 11 for US-UMB. The m0 simulation has the coolest daytime and warmest
nighttime leaf temperatures. Inclusion of a turbulence closure (m1) warms daytime temperatures
and cools nighttime temperatures. The leaf biophysics (b4) reduces the m1 temperature changes,
and the RSL simulations (r1 and r2) further reduce the changes.

Wind speed and temperature profiles simulated with the RSL parameterization are

noticeably different compared with MOST profiles, as shown in Figure 12 for US-UMB. At mid-
day, wind speed in the upper canopy is markedly lower than for MOST, but whereas wind speed
goes to zero with MOST, the RSL wind speed remains finite. Mid-day MOST air temperature in
the canopy increases monotonically to a maximum of 28.5 °C, but the RSL produces a more
complex profile with a temperature maximum of about 26.5 °C in the mid-canopy and lower
temperatures near the ground. During the night, the upper canopy cools to a temperature of about
15 °C, but temperatures in the lower canopy remain warm. The other forest sites show similar
profiles.

**4 Discussion**



The multi-layer canopy with the RSL (ML+RSL) improves the simulation of surface fluxes
compared to the CLM4.5 at most forest and herbaceous sites (Table 5). In terms of $\lambda E$, the
turbulence closure using the CLM4.5 MOST above the canopy and a mixing length model in the
canopy (with $\eta = 3$) substantially reduces RMSE compared to the well-mixed assumption in
which the canopy has the same temperature, water vapor concentration, and wind speed as the
reference height (m0, m1; Figure 7). A similar result is seen for $H$ (Figure 8). This finding is
consistent with Juang et al. (2008), who showed that first-order turbulence closure improves
simulations in a multi-layer canopy compared with the well-mixed assumption.

Additional improvement in $\lambda E$ comes from the leaf biophysics (particularly stomatal

conductance and photosynthetic capacity) (b1, b2; Figure 7). This is consistent with Bonan et al.
(2014), who previously showed improvements arising from the multi-layer canopy, stomatal
conductance, and photosynthetic capacity at the forest sites. Differences between the CLM4.5
and ML+RSL stomatal models likely reflects differences in parameters (slope $g_1$ for CLM4.5;
marginal water-use efficiency $\iota$ for ML+RSL) rather than model structure (Franks et al., 2017).
Further differences arise from the plant hydraulics (Bonan et al., 2014). The RSL has
comparatively little effect on $\lambda E$ (r1, r2; Figure 7). $H$ is similarly improved by the leaf
biophysics, but is degraded by the RSL (Figure 8) because of an increase in the peak mid-day
flux. Harman (2012) also found that the RSL has negligible effect on $\lambda E$ because this flux is
dominated by stomatal conductance, but increases the peak $H$.

The influence of the RSL is evident in the improved relationship between $H$ and the

surface–air temperature difference ($T_{rad} - T_{ref}$) at forest sites (Figure 6). In the CLM4.5, a larger
temperature difference is needed to produce the same positive heat flux to the atmosphere
compared with the observations. With the RSL, a smaller temperature difference gives the same





sensible heat flux, comparable to the observations. This is expected from the RSL theory because
of the larger aerodynamic conductance. Similar such improvement is not seen at the crop site
(US-ARM) because the measurements were taken above the RSL.

The influence of the RSL is also evident in nighttime $u_*$ (Figure 4). Substantial reduction

in RMSE is seen in the m1 simulation (Figure 9), which closely mimics the CLM4.5 in terms of
leaf biophysics and use of MOST above the canopy. The different numerical methods used
between the multi-layer canopy and the CLM4.5 to solve for canopy temperature, surface fluxes,
and the Obukhov length may explain the poor CLM4.5 simulations. The RSL parameterization
further improves $u_*$ (r1, r2; Figure 9), primarily by increasing $u_*$ at night.

Another outcome of the RSL in seen in $T_{rad}$ and leaf temperature. The lowest RMSE

occurs with the well-mixed approximation (m0; Figure 10), which also produces the coolest
daytime and warmest nighttime leaf temperatures (m0; Figure 11). Adding a turbulence closure
(m1) substantially warms daytime leaf temperatures and cools nighttime temperatures, which
degrades the $T_{rad}$ RMSE. The RSL (r1, r2) decreases the daytime temperatures and warms the
nighttime temperatures, which improve the RMSE. Leaf temperatures are cooler during the day
and warmer at night compared with the CLM4.5. Overall, the diurnal temperature range
improves in the ML+RSL simulation compared to that from the CLM4.5, seen in both the
nighttime minimum and the daytime maximum of $T_{rad}$ (Figure 4). This latter improvement is
particularly important given the use of radiometric land surface temperature as an indicator of the
climate impacts of land cover change (Alkama and Cescatti, 2016).

The simulation of wind and temperature profiles is a key outcome of the multi-layer

canopy and RSL. During the day, the CLM4.5 simulates a warmer canopy air space than the
ML+RSL simulation (Figure 12). Air temperature obtained from MOST increases monotonically



towards the bulk surface, whereas the ML+RSL simulation produces a more complex vertical
profile with a maximum located in the upper canopy and cooler temperatures in the lower
canopy. Geiger (1927) first described such profiles, seen also in some studies (Jarvis and
McNaughton, 1986; Pyles et al., 2000; Staudt et al., 2011). The simulated nighttime temperatures
are warmer than the CLM4.5. Temperature profiles have a minimum in the upper canopy, above
which temperature increases with height. However, temperatures increase in the lower canopy.
Nighttime temperatures in a walnut orchard show a minimum in the upper canopy arising from
radiative cooling, but the temperature profile in the lower canopy is more uniform than seen in
Figure 12 (Patton et al., 2011). Enhanced diffusivity resulting from convective instability in the
canopy makes the temperature profile more uniform in the Patton et al. (2011) observations; this
process is lacking in the RSL parameterization. Ryder et al. (2016) and Chen et al. (2016) noted
the difficulty in modeling nighttime temperature profiles in forests and introduced in
ORCHIDEE-CAN an empirical scaling factor to $K_c$ that varies over the day. The results of the
present study, too, suggest that turbulent mixing in conditions where the stratification within and
above the canopy differ in sign needs additional consideration. The importance of within-canopy
temperature gradients is seen in that the microclimatic influence of dense forest canopies buffers
the impact of macroclimatic warming on understory plants (De Frenne et al., 2013) and the
vertical climatic gradients in tropical rainforests are steeper than elevation or latitudinal gradients
(Scheffers et al., 2013).

Various ad hoc changes have been introduced into the next version of the Community

Land Model (CLM5) to correct the deficiencies in $u_*$ and $T_{rad}$. In particular, the Monin–
Obukhov stability parameter has been constrained in stable conditions so that $(z - d)/L_{MO} \leq 0.5$.
This change increases nighttime $u_*$, increases sensible heat transfer to the surface at night, and





increases nighttime $T_{rad}$ (not shown). In contrast, the ML+RSL simulation reduces these same
biases, but resulting from a clear theoretical basis describing canopy-induced physics.

The canopy model encapsulates conservation equations for $\theta$ and $q$, the energy balance

for the sunlit and shaded canopy, and the ground surface energy balance. The various terms in
Eqs. (16) and (17), the governing equations, are easily derived from flux equations and relate to
the leaf ($g_b$, $g_{\ell sun}$, $g_{\ell sha}$) and aerodynamic ($g_a$) conductances, leaf and canopy air storage terms
($c_L$, $\rho_m \Delta z / \Delta t$), plant area index and the sunlit fraction ($\Delta L$, $f_{sun}$), net radiation ($R_{n\ell sun}$, $R_{n\ell sha}$),
and soil surface ($R_{n0}$, $h_{s0}$, $g_{s0}$, $\kappa_{soil}$, $T_{soil}$). These are all terms that need to be defined in land
surface models (except for the storage terms which are commonly neglected), and so the only
new term introduced into the flux equations is leaf heat capacity, but that is obtained from the
leaf mass per area, which is a required parameter in the CLM4.5.

The Harman and Finnigan (2007, 2008) RSL parameterization provides the necessary

aerodynamic conductances and wind speed. It produces a comparable representation of surface-
atmosphere exchange of heat, water and carbon, including within-canopy exchange, to those
based on Lagrangian dynamics (e.g., McNaughton and van den Hurk, 1995) and localized near-
field theory (e.g., Raupach, 1989; Raupach et al., 1997; Siqueira et al., 2003; Ryder et al., 2016;
Chen et al., 2016). Lagrangian representations have the advantage in that they retain closer
fidelity to the underlying dynamics governing exchange. In contrast, however, the RSL
formulation provides linked representations for both momentum and (passive) scalar exchange.
This coupling, impossible with Lagrangian formulations as there is no locally-conserved
equivalent quantity to scalar concentration for momentum, reduces the degrees of freedom
involved. The RSL's linked formulation also facilitates the propagation of knowledge about the
transport of one quantity onto the transport of all other quantities considered. Unlike Lagrangian





formulations, the RSL formulation also naturally asymptotes towards the standard surface layer
representations as required, e.g., with increasing height above ground or for short canopies.

Furthermore, the components of the RSL formulation are far easier to observe than those

in the Lagrangian representations. In particular, the vertical profile of the Lagrangian time scale
($T_L$), critical to the localized near-field formulation, is extremely difficult to determine from
observations or higher-order numerical simulations. Most understanding around $T_L$ is indirect,
heuristic, or tied to an inverted model (Massman and Weil, 1999; Haverd et al., 2009). Finally, it
is worth noting that the RSL formulation is derived from the scales of the coherent and dominant
turbulent structures and directly incorporates canopy architecture (Raupach et al., 1996; Finnigan
et al., 2009), thereby permitting future adaptation of the formulation to advances in our
understanding of the structure and role of turbulence, e.g. to variation with canopy architecture,
landscape heterogeneity, or in low wind conditions.  Far greater effort would be required to
update the parameterizations of the components in the Lagrangian representations to advances in
the understanding of turbulence.

The Harman and Finnigan (2007, 2008) RSL parameterization eliminates a priori

specification of roughness length and displacement height, but introduces other parameters.
Critical parameters are the drag coefficient of canopy elements in each layer ($c_d = 0.25$), the
value of $u_* / u(h)$ for neutral conditions ($\beta_N = 0.35$), and the Schmidt number at the canopy top
with a nominal value $S_c = 0.5$ as modified for atmospheric stability using Eq. (54). These
parameters have physical meaning, are largely observable, have a well-defined range of observed
values, and are not unconstrained parameters to fit the model to observations. The expressions
for $\beta$ and $S_c$ given by Eqs. (51) and (54) are observationally-based, but nevertheless are
heuristic (Harman and Finnigan, 2007, 2008). The parameter $c_2$ relates to the depth scale of the





RSL and though $c_2$ can have complex expressions, a simplification is to take $c_2 = 0.5$ (Harman
and Finnigan, 2007, 2008; Harman, 2012).

The plant canopies simulated in this study are dense canopies in the sense that most of the

momentum is absorbed by plant elements. Appendix A4 provides a modification for sparse
canopies (e.g., plant area index $< 1$ m$^2$ m$^{-2}$) whereby $\beta$ decreases, but this extension to sparse
canopies is largely untested. Raupach (1994) and Massman (1997) also decrease $\beta$ with sparse
canopies. We note that the same challenge occurs in land surface models such as the CLM4.5,
with parameterizations to account for the effects of canopy denseness on within-canopy
turbulence (Zeng et al., 2005).

The RSL parameterization has limits to its applicability; $L_c / L$ must be greater than some

critical value related to $\beta$ in unstable conditions and less than some critical value in stable
conditions (Harman and Finnigan, 2007). We constrained $\beta$ to a value between 0.5 (unstable)
and 0.2 (stable). In practice, this means that $L_c / L \geq -0.79$ (unstable) and $L_c / L \leq 3.75$ (stable),
which satisfies the theoretical limits given by Harman and Finnigan (2007). This range of values
for $\beta$ is consistent with observations above forest canopies shown in Harman and Finnigan
(2007) and is comparable with other parameterizations. Data presented by Raupach (1994) show
a similar range in $\beta$ for full plant canopies, and his parameterization has a maximum value of
0.3. Massman's (1997) parameterization of $\beta$ has a maximum value of 0.32 for full canopies,
but he notes that other studies suggest a range of 0.15–0.25 to 0.40. The Harman and Finnigan
(2007) parameterization used here has the advantage of being consistent with current RSL theory
(Raupach et al., 1996; Finnigan et al., 2009) and incorporates stability dependence through $\beta$, in
contrast with Raupach (1994) and Massman (1997). Removing the lower limit $\beta \geq 0.2$ has little





effect on the simulations, while the upper limit $\beta \leq 0.5$ acts to suppress daytime $u_*$ at some sites
(not shown).
$l_m / \beta$ is a critical length scale in the RSL theory. It modifies flux–profile relationships
($\hat{\phi}_m$, $\hat{\phi}_c$) and also the profiles for $u$ and $K_c$ in the canopy given by Eqs. (21) and (22). These
latter profiles decline exponentially with greater depth in the canopy in relation to $l_m / \beta$, which
can be equivalently written as $0.5 c_d a / \beta^2$ substituting $l_m$ from Eq. (55) and $L_c$ from Eq. (56).
For a particular canopy defined by $c_d$ and $a = (L_T + S_T) / h$, the exponential within-canopy
profile is bounded by the limits placed on $\beta$. Further insight is gained from an equivalent form
of the wind profile equation in which $u(z) = u(h) \exp[-\eta(1 - z / h)]$ with $\eta = h\beta / l_m$. A typical
value of $\eta$ reported in observational studies is 2–4 (Thom, 1975; Cionco, 1978; Brutsaert, 1982).
Comparing equations shows that $\eta = 0.5 c_d (L_T + S_T) / \beta^2$. The constraint $0.2 \leq \beta \leq 0.5$ places
limits to $\eta$. The maximum plant area index in our simulations is 7.2 $m^2$ $m^{-2}$ at US-Dk2. With
$c_d = 0.25$, $\eta$ has values from 3.6 to 22.5. This allows for quite low wind speed and conductance
within the canopy. Diabatic stability within the canopy can differ from that above the canopy.
This would be reflected in the wind speeds used to calculate the leaf conductances and also the
conductance network used to calculate within canopy scalar profiles. For these reasons, we
employ minimum values to the within-canopy wind speed and aerodynamic conductances.

**5 Conclusion**
For over 30 years, land surface models have parameterized surface fluxes using a dual-source
canopy in which the vegetation is treated as a big-leaf without vertical structure and in which
MOST is used to parameterize turbulent fluxes above the canopy. The RSL parameterization of



Harman and Finnigan (2007, 2008) provides a means to represent turbulent processes extending
from the ground through the canopy and the RSL with sound theoretical underpinnings of
canopy-induced turbulence and with few additional parameters. The implementation of the RSL
improves model performance in terms of sensible heat flux, friction velocity, and radiative
temperature, and additional improvement comes from advances in modeling stomatal
conductance and canopy physiology beyond what is in the CLM4.5. Indeed, the modeling of
canopy turbulence and canopy physiology are inextricably linked (Finnigan and Raupach 1987),
and the 30+ years of land surface models has likely lead to compensating insufficiency in both.

Multi-layer canopies are becoming practical for land surface models, seen in the

ORCHIDEE-CAN model (Ryder et al., 2016; Chen et al., 2016) and in this study. A multi-layer
canopy facilitates the treatment of plant hydraulic control of stomatal conductance (Williams et
al., 1996; Bonan et al., 2014), provides new ways to test models directly with leaf-level
measurements in the canopy, and is similar to the canopy representations used in canopy-
chemistry models (Stroud et al., 2005; Forkel et al., 2006; Wolfe and Thornton, 2011; Ashworth
et al., 2015). Here, we provide a tractable means to simulate the necessary profiles of wind
speed, temperature, and water vapor. While this is an advancement over the CLM4.5, much work
remains to fully develop this class of model. Significant questions remain about how well multi-
layer models capture the profiles of air temperature, water vapor, and leaf temperature in the
canopy, how important these profiles are for vegetation source fluxes, and how many canopy
layers are needed to adequately represent gradients in the canopy. The testing of ORCHIDEE-
CAN (Chen et al., 2016) has begun to address these questions, but high quality measurements in
canopies are required to better distinguish among turbulence parameterizations (e.g., Patton et
al., 2011). Moreover, multi-layer canopies raise a fundamental question about the interface



between the atmosphere and land surface. The coupling of the Community Land Model with the
atmosphere depicts the land as a bulk source/sink for heat, moisture, and momentum, and these
fluxes are boundary conditions to the atmosphere model. Multi-layer canopy models simulate a
volume of air extending from some level in the atmosphere to the ground. A critical question that
remains unresolved is where does the parameterization of the atmospheric boundary layer stop
and the land surface model begin.

**Code availability**
The multi-layer canopy runs independent of the CLM4.5, but utilizes common code (e.g., soil
temperature). The canopy flux code is available at https://github.com/gbonan/CLM-ml_v0.

**Appendix A: Model description**
**A1 Derivation of Eqs. (16) and (17)**
Eq. (10) for the energy balance of the sunlit portion of layer $i$ can be algebraically rewritten as
$$T_{\ell sun,i}^{n+1} = \alpha_i^{sun} \theta_i^{n+1} + \beta_i^{sun} q_i^{n+1} + \delta_i^{sun} \tag{30}$$
with
$$\alpha_i^{sun} = \frac{2 c_p g_{b,i}}{2 c_p g_{b,i} + \lambda s_i^{sun} g_{\ell sun,i} + c_{L,i} / \Delta t} \tag{31}$$
$$\beta_i^{sun} = \frac{\lambda g_{\ell sun,i}}{2 c_p g_{b,i} + \lambda s_i^{sun} g_{\ell sun,i} + c_{L,i} / \Delta t} \tag{32}$$
$$\delta_i^{sun} = \frac{R_{n\ell sun,i} - \lambda \left[ q_{sat}\left(T_{\ell sun,i}^n\right) - s_i^{sun} T_{\ell sun,i}^n \right] g_{\ell sun,i} + c_{L,i} T_{\ell sun,i}^n / \Delta t}{2 c_p g_{b,i} + \lambda s_i^{sun} g_{\ell sun,i} + c_{L,i} / \Delta t} \tag{33}$$
Similar coefficients are found from Eq. (13) for the shaded leaf to give


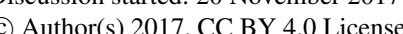


$\qquad T_{\ell sha,i}^{n+1} = \alpha_i^{sha}\theta_i^{n+1} + \beta_i^{sha}q_i^{n+1} + \delta_i^{sha}$ (34)
Eq. (14) for the ground surface energy balance is similarly rewritten as
$\qquad T_0^{n+1} = \alpha_0\theta_1^{n+1} + \beta_0 q_1^{n+1} + \delta_0$ (35)
with
$\qquad \alpha_0 = \dfrac{c_p g_{a,0}}{c_p g_{a,0} + \lambda h_{s0} s_0 g_{s0} + \kappa_{soil}/\Delta z_{soil}}$ (36)
$\qquad \beta_0 = \dfrac{\lambda g_{s0}}{c_p g_{a,0} + \lambda h_{s0} s_0 g_{s0} + \kappa_{soil}/\Delta z_{soil}}$ (37)
$\qquad \delta_0 = \dfrac{R_{n0} - \lambda h_{s0}\left[q_{sat}\left(T_0^n\right) - s_0 T_0^n\right]g_{s0} + T_{soil}^n\kappa_{soil}/\Delta z_{soil}}{c_p g_{a,0} + \lambda h_{s0} s_0 g_{s0} + \kappa_{soil}/\Delta z_{soil}}$ (38)
With these substitutions, Eqs. (8) and (9) are rewritten as Eqs. (16) and (17) with the algebraic
coefficients in Sect. S2 of the Supplement.

**A2 Roughness sublayer parameterization**
The flux–gradient relationships used with Monin–Obukhov similarity theory are
$\qquad \phi_m\left(\zeta\right) = \begin{cases} \left(1-16\zeta\right)^{-1/4} & \zeta < 0 \text{ (unstable)} \\ 1+5\zeta & \zeta \geq 0 \text{ (stable)} \end{cases}$ (39)
for momentum, and
$\qquad \phi_c\left(\zeta\right) = \begin{cases} \left(1-16\zeta\right)^{-1/2} & \zeta < 0 \text{ (unstable)} \\ 1+5\zeta & \zeta \geq 0 \text{ (stable)} \end{cases}$ (40)
for heat and water vapor. These relationships use the dimensionless parameter $\zeta = (z-d)/L_{MO}$.
The integrated similarity functions are

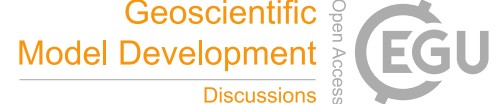



$\quad \psi_m(\zeta) = \begin{cases} 2\ln\left(\dfrac{1+x}{2}\right) + \ln\left(\dfrac{1+x^2}{2}\right) - 2\tan^{-1}x + \dfrac{\pi}{2} & \zeta < 0 \text{ (unstable)} \\ -5\zeta & \zeta \geq 0 \text{ (stable)} \end{cases}$ (41)
$\quad$ with $x = (1-16\zeta)^{1/4}$, and
$\quad \psi_c(\zeta) = \begin{cases} 2\ln\left(\dfrac{1+x^2}{2}\right) & \zeta < 0 \text{ (unstable)} \\ -5\zeta & \zeta \geq 0 \text{ (stable)} \end{cases}$ (42)
$\quad$ These equations are valid for moderate values of $\zeta$ from about –2 to 1 (Foken 2006), and we
$\quad$ adopt a similar restriction.
$\qquad$ The RSL parameterization modifies Monin–Obukhov similarity theory by introducing an
$\quad$ additional dimensionless parameter $\xi = (z-d)\beta/l_m$, which is the height $z-d$ normalized by
$\quad$ the length scale $l_m/\beta$. In Harman and Finnigan (2007, 2008), the modified flux–gradient
$\quad$ relationship for momentum is
$\quad \Phi_m(z) = \phi_m\left(\dfrac{z-d}{L_{MO}}\right)\hat{\phi}_m\left(\dfrac{z-d}{l_m/\beta}\right)$ (43)
$\quad$ with
$\quad \hat{\phi}_m(\xi) = 1 - c_1\exp(-c_2\xi)$ (44)
$\quad$ and
$\quad c_1 = \left[1 - \dfrac{k}{2\beta}\phi_m^{-1}\left(\dfrac{h-d}{L_{MO}}\right)\right]\exp(c_2/2)$ (45)
$\quad$ and a simplification is to take $c_2 = 0.5$. The integrated RSL function $\hat{\psi}_m$ is
$\quad \hat{\psi}_m(z) = \displaystyle\int_{z-d}^{\infty} \phi_m\left(\dfrac{z'}{L_{MO}}\right)\left[1 - \hat{\phi}_m\left(\dfrac{z'}{l_m/\beta}\right)\right]\dfrac{dz'}{z'}$ (46)





For scalars, the flux–gradient relationship in Harman and Finnigan (2008) is
$$\Phi_c(z) = \phi_c\left(\frac{z-d}{L_{MO}}\right)\hat{\phi}_c\left(\frac{z-d}{l_m/\beta}\right) \tag{47}$$
The RSL function $\hat{\phi}_c$ is evaluated the same as for $\hat{\phi}_m$ using Eq. (44), but with
$$c_1 = \left[1 - \frac{S_c k}{2\beta}\phi_c^{-1}\left(\frac{h-d}{L_{MO}}\right)\right]\exp(c_2/2) \tag{48}$$
$\hat{\psi}_c$ is evaluated similar to $\hat{\psi}_m$ using Eq. (46), but with $\phi_c$ and $\hat{\phi}_c$.

The functions $\hat{\psi}_m$ and $\hat{\psi}_c$ must be integrated using numerical methods. In practice,

however, values can be obtained from a look-up table. Eq. (46) can be expanded using Eq. (44)
for $\hat{\phi}_m$ and using $l_m/\beta = 2(h-d)$ from Eq. (57) so that an equivalent equation is
$$\hat{\psi}_m(z) = c_1 \int_{z-d}^{\infty} \phi_m\left(\frac{z'}{L_{MO}}\right)\exp\left[-\frac{c_2 z'}{2(h-d)}\right]\frac{dz'}{z'} \tag{49}$$
The lower limit of integration in Eq. (49) can be rewritten as $z-d=(z-h)+(h-d)$ and
dividing both sides by $h-d$ gives the expression $(z-h)/(h-d)+1$. In this notation, Eq. (49)
becomes
$$\hat{\psi}_m(z) = c_1 \int_{\frac{z-h}{h-d}+1}^{\infty} \phi_m\left[\frac{(h-d)z'}{L_{MO}}\right]\exp\left(-\frac{c_2 z'}{2}\right)\frac{dz'}{z'} \tag{50}$$
In this equation, the integral is specified in a non-dimensional form and depends on two non-
dimensional parameters: $(z-h)/(h-d)$ and $(h-d)/L_{MO}$. The integral is provided in a look-up
table as $A[(z-h)/(h-d),(h-d)/L_{MO}]$. $\hat{\psi}_m$ is then given by $c_1 A$. A similar approach gives $\hat{\psi}_c$.

An expression for $\beta$ is obtained from the relationship





$\qquad \beta \phi_m \left( \beta^2 L_c / L_{MO} \right) = \beta_N$ (51)
with $\beta_N$ the value of $u_* / u(h)$ for neutral conditions (a representative value is $\beta_N = 0.35$, which
is used here). Using Eq. (39) for $\phi_m$, the expanded form of Eq. (51) for unstable conditions
($L_{MO} < 0$) is a quadratic equation for $\beta^2$ given by
$\qquad \left( \beta^2 \right)^2 + 16 \dfrac{L_c}{L_{MO}} \beta_N^4 \left( \beta^2 \right) - \beta_N^4 = 0$ (52)
The correct solution is larger of the two roots. For stable conditions ($L_{MO} > 0$), a cubic equation
is obtained for $\beta$ whereby
$\qquad 5 \dfrac{L_c}{L_{MO}} \beta^3 + \beta - \beta_N = 0$ (53)
This equation has one real root. We restrict $\beta$ to be in the range 0.2–0.5 (see Discussion for
further details).
$\qquad$ The Schmidt number ($S_c$) is parameterized by Harman and Finnigan (2008) as
$\qquad S_c = 0.5 + 0.3 \tanh \left( 2 L_c / L_{MO} \right)$ (54)
$\qquad$ Eq. (21) is derived from the momentum balance equation with a first-order turbulence
closure in which the eddy diffusivity is specified in relation to a mixing length ($l_m$) that is
constant with height. From this, Harman and Finnigan (2007) obtained expressions for $l_m$ and $d$
so that
$\qquad l_m = 2 \beta^3 L_c$ (55)
with
$\qquad L_c = \left( c_d a \right)^{-1}$ (56)
and

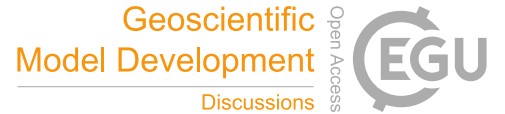

$$h - d = \frac{l_m}{2\beta} = \beta^2 L_c \qquad (57)$$
The term $L_c$ is the canopy length scale (m), specified by the dimensionless leaf aerodynamic
drag coefficient (a common value is $c_d = 0.25$, which is used here) and plant area density ($a$, m²
m⁻³). For Eq. (56), plant area density is estimated as the leaf and stem area index ($L_T + S_T$)
divided by canopy height ($h$).

**A3 Obukhov length**
The Obukhov length is
$$L_{MO} = \frac{u_*^2 \theta_{vref}}{kg \theta_{v*}} \qquad (58)$$
with $\theta_{vref}$ the virtual potential temperature (K) at the reference height, and $\theta_{v*}$ the virtual
potential temperature scale (K) given as
$$\theta_{v*} = \theta_* + 0.61\theta_{ref} q_{*,kg} \qquad (59)$$
The solution to $L_{MO}$ requires an iterative numerical calculation (Figure 2). A value for $\beta$ is
obtained for an initial estimate of $L_{MO}$ using Eq. (51), which gives the displacement height ($d$)
using Eq. (57). The Schmidt number ($S_c$) is calculated for the current $L_{MO}$ using Eq. (54). The
functions $\phi_m$ and $\phi_c$ are evaluated using Eqs. (39) and (40) at the canopy height ($h$) to obtain the
parameter $c_1$ as in Eqs. (45) and (48). The similarity functions $\psi_m$ and $\psi_c$ are evaluated at $z$
and $h$ using Eqs. (41) and (42). The RSL functions $\hat{\psi}_m$ and $\hat{\psi}_c$ are evaluated at $z$ and $h$ from a
look-up table. $u_*$ is obtained from Eq. (19) using the wind speed ($u_{ref}$) at the reference height
($z_{ref}$). $\theta_*$ is calculated from Eq. (20) using $\theta_{ref}$ for the current timestep and $\theta(h)$ for the previous



sub-timestep, and a comparable equation provides $q_*$. A new estimate of $L_{MO}$ is obtained, and
the iteration is repeated until convergence in $L_{MO}$ is achieved.

**A4 Sparse canopies**
The RSL theory of Harman and Finnigan (2007, 2008) was developed for dense canopies. Sparse
canopies can be represented by adjusting $\beta_N$, $d$, and $S_c$ for plant area index ($L_T + S_T$). The
neutral value for $\beta$ is
$$\beta_N = \left[ c_\beta + 0.3 \left( L_T + S_T \right) \right]^{1/2} \le \beta_{N\max} \tag{60}$$
where
$$c_\beta = k^2 \left[ \ln \left( \frac{h + z_{0m}}{z_{0m}} \right) \right]^{-2} \tag{61}$$
and $z_{0m} = 0.01$ m is the roughness length for momentum of the underlying ground surface. $\beta_N$
is constrained to be less than a maximum value for neutral conditions ($\beta_{N\max} = 0.35$). The
displacement height is
$$h - d = \beta^2 L_c \left\{ 1 - \exp \left[ -0.25 \left( L_T + S_T \right) / \beta^2 \right] \right\} \tag{62}$$
The Schmidt number is
$$S_c = \left( 1 - \frac{\beta_N}{\beta_{N\max}} \right) 1.0 + \frac{\beta_N}{\beta_{N\max}} \left[ 0.5 + 0.3 \tanh \left( 2 L_c / L_{MO} \right) \right] \tag{63}$$
This equation weights the Schmidt number between that for a neutral surface layer (1.0) and the
RSL value calculated from Eq. (54).

**Appendix B: List of symbols, their definition, and units**





| Symbol | Description |
| --- | --- |
| $a_i$ | Plant area density ($m^2$ $m^{-3}$) |
| $A_n$ | Leaf net assimilation ($\mu$mol $CO_2$ $m^{-2}$ $s^{-1}$) |
| $c_1$, $c_2$ | Scaled magnitude ($c_1$) and height ($c_2 = 0.5$), respectively, for the RSL functions (–) |
| $c_d$ | Leaf aerodynamic drag coefficient (0.25) |
| $c_{dry}$ | Specific heat of dry biomass (1396 J $kg^{-1}$ $K^{-1}$) |
| $c_{L,i}$ | Heat capacity of leaves (J $m^{-2}$ leaf area $K^{-1}$) |
| $c_p$ | Specific heat of air, $c_{pd}(1+0.84q_{ref.kg})M_d$ (J $mol^{-1}$ $K^{-1}$) |
| $c_{pd}$ | Specific heat of dry air at constant pressure (1005 J $kg^{-1}$ $K^{-1}$) |
| $c_s$ | Leaf surface $CO_2$ concentration ($\mu$mol $mol^{-1}$) |
| $c_v$ | Soil heat capacity (J $m^{-3}$ $K^{-1}$) |
| $c_{wat}$ | Specific heat of water (4188 J $kg^{-1}$ $K^{-1}$) |
| $c_\beta$ | Parameter for $\beta_N$ in sparse canopies (–) |
| $d$ | Displacement height (m) |
| $e_{ref}$ | Reference height vapor pressure (Pa) |
| $E_i$ | Water vapor flux (mol $H_2O$ $m^{-2}$ $s^{-1}$) |
| $E_0$ | Soil evaporation (mol $H_2O$ $m^{-2}$ $s^{-1}$) |
| $E_{\ell sun,i}$, $E_{\ell sha,i}$ | Evaporative flux for sunlit or shaded leaves (mol $H_2O$ $m^{-2}$ plant area $s^{-1}$) |
| $f_c$ | Carbon content of dry biomass (0.5 g C $g^{-1}$) |



| | |
|---|---|
| $f_{dry,i}$ | Dry transpiring fraction of canopy (–) |
| $f_{green,i}$ | Green fraction of canopy (–) |
| $f_i$ | Leaf nitrogen relative to canopy top (–) |
| $f_{sun,i}$ | Sunlit fraction of canopy (–) |
| $f_w$ | Water content of fresh biomass (0.7 g $H_2O$ g$^{-1}$) |
| $f_{wet,i}$ | Wet fraction of canopy (–) |
| $g$ | Gravitational acceleration (9.80665 m s$^{-2}$) |
| $g_0$, $g_1$ | Intercept (mol $H_2O$ m$^{-2}$ s$^{-1}$) and slope (–) for Ball–Berry stomatal conductance |
| $g_{a,i}$ | Aerodynamic conductance (mol m$^{-2}$ s$^{-1}$) |
| $g_{b,i}$ | Leaf boundary layer conductance (mol m$^{-2}$ s$^{-1}$) |
| $g_{\ell sun,i}$, $g_{\ell sha,i}$ | Leaf conductance for sunlit or shaded leaves (mol $H_2O$ m$^{-2}$ s$^{-1}$) |
| $g_s$ | Stomatal conductance (mol $H_2O$ m$^{-2}$ s$^{-1}$); $g_{sun,i}$, sunlit leaves; $g_{sha,i}$, shaded leaves |
| $g_{s0}$ | Total surface conductance for water vapor (mol $H_2O$ m$^{-2}$ s$^{-1}$) |
| $g_{soil}$ | Soil conductance for water vapor (mol $H_2O$ m$^{-2}$ s$^{-1}$) |
| $G_0$ | Soil heat flux (W m$^{-2}$) |
| $h$ | Canopy height (m) |
| $h_s$ | Fractional relative humidity at the leaf surface (–) |
| $h_{s0}$ | Fractional relative humidity at the soil surface (–) |
| $H_i$ | Sensible heat flux (W m$^{-2}$) |





| | |
|---|---|
| $H_0$ | Soil sensible heat flux (W m$^{-2}$) |
| $H_{\ell sun,i}$ , $H_{\ell sha,i}$ | Sensible heat flux for sunlit or shaded leaves (W m$^{-2}$ plant area) |
| $i$ | Canopy layer index |
| $k$ | von Karman constant (0.4) |
| $K_{c,i}$ | Scalar diffusivity (m$^2$ s$^{-1}$) |
| $K_n$ | Canopy nitrogen decay coefficient (–) |
| $l_m$ | Mixing length for momentum (m) |
| $L_c$ | Canopy length scale (m) |
| $L_{MO}$ | Obukhov length (m) |
| $L_T$ | Canopy leaf area index (m$^2$ m$^{-2}$) |
| $\Delta L_i$ | Canopy layer plant area index (m$^2$ m$^{-2}$) |
| $\Delta L_{sun,i}$ , $\Delta L_{sha,i}$ | Plant area index of sunlit or shaded canopy layer (m$^2$ m$^{-2}$) |
| $\bar{M}$ | Molecular mass of moist air, $\rho / \rho_m$ (kg mol$^{-1}$) |
| $M_a$ | Leaf carbon mass per unit area (g C m$^{-2}$ leaf area) |
| $M_d$ | Molecular mass of dry air (0.02897 kg mol$^{-1}$) |
| $M_w$ | Molecular mass of water (0.01802 kg mol$^{-1}$) |
| $n$ | Time index (–) |
| $P_{ref}$ | Reference height air pressure (Pa) |
| $q_i$ | Water vapor concentration (mol mol$^{-1}$) |
| $q_0$ | Soil surface water vapor concentration (mol mol$^{-1}$) |



| | |
|---|---|
| $q_{ref}$ | Reference height water vapor concentration (mol mol$^{-1}$) |
| $q_{ref.kg}$ | Reference height specific humidity, $0.622e_{ref}/(P_{ref}-0.378e_{ref})$ (kg kg$^{-1}$) |
| $q_{sat}(T)$ | Saturation water vapor concentration (mol mol$^{-1}$) at temperature $T$ |
| $q_*$ | Characteristic water vapor scale (mol mol$^{-1}$) |
| $q_{*.kg}$ | Characteristic water vapor scale, $q_* M_w/\bar{M}$ (kg kg$^{-1}$) |
| $R_{n0}$ | Soil surface net radiation (W m$^{-2}$) |
| $R_{n\ell sun,i}$, $R_{n\ell sha,i}$ | Net radiation for sunlit or shaded leaves (W m$^{-2}$ plant area) |
| $\Re$ | Universal gas constant (8.31446 J K$^{-1}$ mol$^{-1}$) |
| $s_i^{sun}$, $s_i^{sha}$ | Temperature derivative of saturation water vapor concentration evaluated at $T_{\ell sun,i}$ and $T_{\ell sha,i}$, $dq_{sat}/dT$ (mol mol$^{-1}$ K$^{-1}$) |
| $s_0$ | Temperature derivative of saturation water vapor concentration evaluated at the soil surface temperature $T_0$, $dq_{sat}/dT$ (mol mol$^{-1}$ K$^{-1}$) |
| $S_c$ | Schmidt number at the canopy top (–) |
| $S_T$ | Canopy stem area index (m$^2$ m$^{-2}$) |
| $t$ | Time (s) |
| $T_0$ | Soil surface temperature (K) |
| $T_{\ell sun,i}$, $T_{\ell sha,i}$ | Temperature of sunlit or shaded leaves (K) |
| $T_{ref}$ | Reference height temperature (K) |
| $T_{soil}$ | Temperature of first soil layer (K) |
| $u_i$ | Wind speed (m s$^{-1}$) |





| | |
|---|---|
| $u_{ref}$ | Reference height wind speed (m s$^{-1}$) |
| $u_*$ | Friction velocity (m s$^{-1}$) |
| $V_{c\max}$ | Maximum carboxylation rate (µmol m$^{-2}$ s$^{-1}$) |
| $W_i$ | Intercepted water (kg H$_2$O m$^{-2}$) |
| $z_i$ | Height (m) |
| $z_{ref}$ | Reference height (m) |
| $z_{0m}$, $z_{0c}$ | Roughness length of ground for momentum (0.01 m) and scalars (0.001 m), respectively |
| $\Delta z_{soil}$ | Depth of first soil layer (m) |
| $\beta$ | Ratio of friction velocity to wind speed at the canopy height (–) |
| $\beta_N$ | Neutral value of $\beta$ (0.35) |
| $\beta_{N\max}$ | Maximum value of $\beta_N$ in a sparse canopy (0.35) |
| $\zeta$ | Monin–Obukhov dimensionless parameter (–) |
| $\theta_i$ | Potential temperature (K) |
| $\theta_{ref}$ | Reference height potential temperature (K) |
| $\theta_s$ | Aerodynamic surface temperature (K) |
| $\theta_{vref}$ | Reference height virtual potential temperature (K) |
| $\theta_{v*}$ | Characteristic virtual potential temperature scale (K) |
| $\theta_*$ | Characteristic potential temperature scale (K) |
| $\iota$ | Marginal water-use efficiency parameter (µmol CO$_2$ mol$^{-1}$ H$_2$O) |
| $\kappa_{soil}$ | Thermal conductivity of first soil layer (W m$^{-1}$ K$^{-1}$) |





| $\xi$ | RSL dimensionless parameter (–) |
|---|---|
| $\lambda$ | Latent heat of vaporization (45.06802 kJ mol$^{-1}$) |
| $\rho$ | Density of moist air, $\rho_m M_d (1 - 0.378 e_{ref} / P_{ref})$ (mol m$^{-3}$) |
| $\rho_m$ | Molar density, $P_{ref} / \Re T_{ref}$ (mol m$^{-3}$) |
| $\phi_m$, $\phi_c$ | Monin–Obukhov similarity theory flux–gradient relationships for momentum and scalars (–) |
| $\hat{\phi}_m$, $\hat{\phi}_c$ | RSL modification of flux–gradient relationships for momentum and scalars (–) |
| $\Phi_m$, $\Phi_c$ | RSL-modified flux–gradient relationships for momentum and scalars (–) |
| $\psi_\ell$, $\psi_{\ell\min}$ | Leaf water potential and its minimum value (MPa) |
| $\psi_m$, $\psi_c$ | Integrated form of Monin–Obukhov stability functions for momentum and scalars (–) |
| $\hat{\psi}_m$, $\hat{\psi}_c$ | Integrated form of the RSL stability functions for momentum and scalars (–) |


**The Supplement related to this article is available online.**

*Author contributions*. E. Patton, I. Harman, and J. Finnigan developed the RSL code. G. Bonan
developed the numerical solution for scalar profiles in the canopy. G. Bonan and E. Patton
implemented the code in the multi-layer canopy. G. Bonan and E. Patton designed the model
simulations. K. Oleson performed the CLM4.5 simulations. Y. Lu provided the US-ARM data,
and E. Burakowski processed the US-Dk1, US-Dk2, and US-Dk3 data. G. Bonan wrote the
manuscript with contributions from all co-authors.



*Competing interests*. The authors declare that they have no conflict of interest.

*Acknowledgments*. The National Center for Atmospheric Research is sponsored by the National
Science Foundation. This work was supported by the National Science Foundation Science and
Technology Center for Multi-Scale Modeling of Atmospheric Processes, managed by Colorado
State University under cooperative agreement No. ATM-0425247.

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




Table 1. Site information for the 4 deciduous broadleaf forest (DBF), 3 evergreen needleleaf
forest (ENF), 2 grassland (GRA), and 3 cropland (CRO) flux towers, including mean annual
temperature (MAT) and annual precipitation (Prec).

| Site | Vegetation type | Latitude | Longitude | MAT (°C) | Prec (mm) | Years | Month | Leaf area index[a] | Canopy height (m) |
|---|---|---|---|---|---|---|---|---|---|
| US-Dk2 | DBF | 35.97 | −79.10 | 14.4 | 1169 | 2004–2008 | July | 6.2 | 25 |
| US-Ha1 | DBF | 42.54 | −72.17 | 6.6 | 1071 | 1992–2006 | July | 4.9 | 23 |
| US-MMS | DBF | 39.32 | −86.41 | 10.8 | 1032 | 1999–2006 | July | 4.7 | 27 |
| US-UMB | DBF | 45.56 | −84.71 | 5.8 | 803 | 1999–2006 | July | 4.2 | 21 |
| US-Dk3 | ENF | 35.98 | −79.09 | 14.4 | 1170 | 2004–2008 | July | 4.7 | 17 |
| US-Ho1 | ENF | 45.20 | −68.74 | 5.3 | 1070 | 1996–2004 | July | 4.6 | 20 |
| US-Me2 | ENF | 44.45 | −121.56 | 6.3 | 523 | 2002–2007 | July | 3.8 | 14 |
| US-Dk1[b] | GRA | 35.97 | −79.09 | 14.4 | 1170 | 2004–2008 | July | 1.7 | 0.5 |

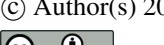



| US-Var | GRA | 38.41 | –120.95 | 15.8 | 559 | 2001–2007 | March | 2.4 | 0.6 |
| US-ARM | CRO | 36.61 | –97.49 | 14.8 | 843 | 2003–4, 2006–7, 2009–10 | April | 2–4 | 0.5 |
| US-Bo1 | CRO | 40.01 | –88.29 | 11.0 | 991 | 1998–2006 (even) | August | 5.0 | 0.9 |
| US-Ne3 | CRO | 41.18 | –96.44 | 10.1 | 784 | 2002, 2004 | August | 3.7 | 0.9 |


[a] Shown is the maximum for the month. Maximum leaf area index for US-ARM varied by year,
and shown is the range in monthly maximum across all years.
[b] $H$ and $u_*$ for 2007 and 2008 are excluded.












Table 2. Leaf heat capacity

| Plant functional type | Specific leaf area $(m^2\ g^{-1}\ C)$ | Leaf mass per area $(g\ dry\ mass\ m^{-2})$ | Heat capacity $(J\ m^{-2}\ K^{-1})$ |
|---|---|---|---|
| Grass, crop | 0.03 | 67 | 745 |
| Deciduous broadleaf tree | 0.03 | 67 | 745 |
| Evergreen needleleaf tree | | | |
|    Temperate | 0.01 | 200 | 2234 |
|    Boreal | 0.008 | 250 | 2792 |







Table 3. Major differences between the CLM4.5 and ML+RSL

| Feature | CLM4.5 | ML+RSL |
|---|---|---|
| Canopy | Dual source: vegetation (sunlit/shaded big-leaf) and soil | Multilayer; sunlit and shaded leaf fluxes at each level; scalar profiles ($u$, $\theta$, $q$) based on conservation equations |
| Plant area index | Big leaf | Vertical profile uses beta distribution probability density function for leaves and uniform profile for stems |
| Stomatal conductance | $g_s = g_0 + g_1 h_s A_n / c_s$ | $\Delta A_n / \Delta E_\ell = \iota$ with $\psi_\ell > \psi_{\ell\min}$; Bonan et al. (2014) |
| Relative leaf nitrogen profile $f_i = \exp[-K_n \sum \Delta L_j]$ | $K_n = 0.3$ | $K_n = \exp(0.00963 V_{c\max} - 2.43)$; Bonan et al. (2014) |
| Storage | – | Plant: $c_L(\Delta T_\ell / \Delta t)$ Air: $\rho_m c_p \Delta z(\Delta \theta / \Delta t)$ Air: $\rho_m \Delta z(\Delta q / \Delta t)$ |
| Above-canopy turbulence | MOST | RSL |
| Within-canopy turbulence | Understory wind speed equals $u_*$; aerodynamic conductance based on $u_*$ and understory $\mathrm{Ri}$. | $u(z) = u(h)\exp\left[(z-h)\beta/l_m\right]$ $K_c(z) = K_c(h)\exp\left[(z-h)\beta/l_m\right]$ |





Table 4. Summary of simulation changes to the turbulence parameterization and leaf biophysics

| Simulation | Turbulence | | Biophysical | | | |
| --- | --- | --- | --- | --- | --- | --- |
| | $\theta$, $q$ | $u$, $g_a$ | $g_s$ | $K_n$ | Plant area density | $c_L$ |
| CLM4.5 | CLM4.5 | CLM4.5 | CLM4.5 | CLM4.5 | $(L_T + S_T)/h$ | – |
| m0 | Well-mixed | – | " | " | " | " |
| m1 | Eqs. (16) and (17) | $z > h$ : CLM4.5 $z < h$ : Eqs. (21) and (26), $\eta = 3$ | " | " | " | " |
| b1 | " | " | Bonan et al. (2014) | " | " | " |
| b2 | " | " | " | Bonan et al. (2014) | " | " |
| b3 | " | " | " | " | Eq. (28) | " |
| b4 | " | " | " | " | " | Eq. (29) |
| r1 | " | $z > h$ : Eqs. (19) and (24) $z < h$ : Eqs. (21) and (26), $\eta = 3$ | " | " | " | " |
| r2 | " | ", but with $l_m / \beta$ | " | " | " | " |






Table 5. Average Taylor skill score for the ML+RSL (first number) and CLM4.5 (second
number) simulations. Skill scores greater than those of CLM4.5 are highlighted in bold.

| Site | $R_n$ | H | $\lambda E$ | $u_*$ | $T_{rad}$ | GPP |
|---|---|---|---|---|---|---|
| Forest | | | | | | |
| US-Ha1 | **0.98**/0.98 | **0.89**/0.85 | **0.94**/0.92 | **0.91**/0.82 | – | **0.83**/0.80 |
| US-MMS | **1.00**/0.99 | 0.44/0.47 | **0.88**/0.87 | **0.84**/0.78 | **0.89**/0.81 | 0.70/0.70 |
| US-UMB | 0.99/0.99 | **0.90**/0.84 | **0.92**/0.88 | **0.93**/0.89 | **0.92**/0.75 | **0.81**/0.73 |
| US-Dk2 | **0.98**/0.98 | **0.53**/0.52 | 0.93/0.93 | **0.86**/0.82 | **0.75**/0.75 | – |
| US-Dk3 | **0.99**/0.99 | **0.85**/0.85 | 0.94/0.94 | 0.81/0.82 | **0.83**/0.79 | – |
| US-Ho1 | 0.96/0.97 | 0.93/0.94 | 0.91/0.93 | **0.92**/0.86 | – | 0.86/0.87 |
| US-Me2 | **1.00**/1.00 | **0.90**/0.79 | **0.89**/0.64 | **0.88**/0.84 | **0.94**/0.78 | **0.91**/0.57 |
| Herbaceous | | | | | | |
| US-Dk1 | 0.99/0.99 | **0.89**/0.87 | 0.90/0.90 | 0.73/0.82 | **0.98**/0.95 | – |
| US-Var | 0.95/0.96 | **0.72**/0.59 | **0.95**/0.95 | **0.81**/0.79 | 0.98/0.98 | **0.89**/0.79 |
| US-Bo1 | 0.99/0.99 | **0.75**/0.61 | **0.96**/0.94 | **0.94**/0.94 | **0.90**/0.85 | – |
| US-Ne3 | **1.00**/1.00 | **0.48**/0.35 | **0.85**/0.77 | **0.98**/0.96 | **0.94**/0.86 | **0.78**/0.59 |
| US-ARM | 0.96/0.97 | **0.93**/0.88 | 0.91/0.94 | 0.95/0.95 | **0.98**/0.97 | – |







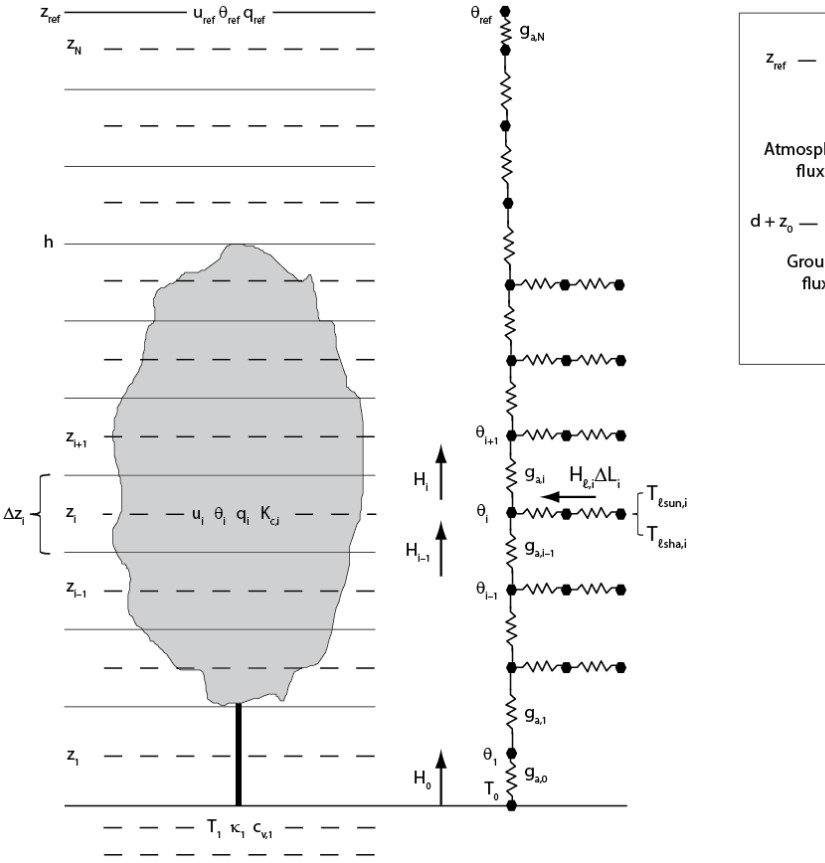

Figure 1. Numerical grid used to represent a multi-layer canopy. The volume of air from the

reference height ($z_{ref}$) to the ground consists of $N$ layers with a thickness $\Delta z_i$, plant area index

$\Delta L_i$, and plant area density $a_i = \Delta L_i / \Delta z_i$. The canopy has a height $h$. Wind speed ($u_i$),

temperature ($\theta_i$), water vapor concentration ($q_i$), and scalar diffusivity ($K_{c,i}$) are physically

centered in each layer at height $z_i$. An aerodynamic conductance ($g_{a,i}$) regulates the turbulent

flux between layer $i$ to $i+1$. The right-hand side of the figure depicts the sensible heat fluxes

below and above layer $i$ ($H_{i-1}$ and $H_i$) and the total vegetation source flux ($H_{\ell,i}\Delta L_i$) with sunlit

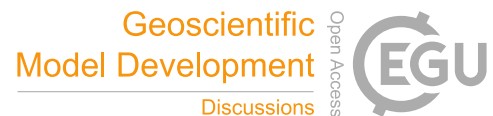

and shaded components. Shown is the conductance network, in which nodal points represent
scalar values in the air and at the leaf. Canopy source fluxes depend on leaf conductances and
leaf temperature, calculated separately for sunlit and shaded leaves using the temperatures $T_{\ell sun,i}$
and $T_{\ell sha,i}$, respectively. The ground is an additional source of heat and water vapor with
temperature $T_0$. The inset panel (a) shows the dual-source canopy model used in the Community
Land Model (CLM4.5). Here, Monin–Obukhov similarity theory provides the flux from the
surface with height $d + z_0$ (displacement height plus roughness length) and temperature $\theta_s$ to the
reference height with the conductance $g_a$.






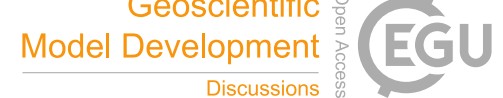



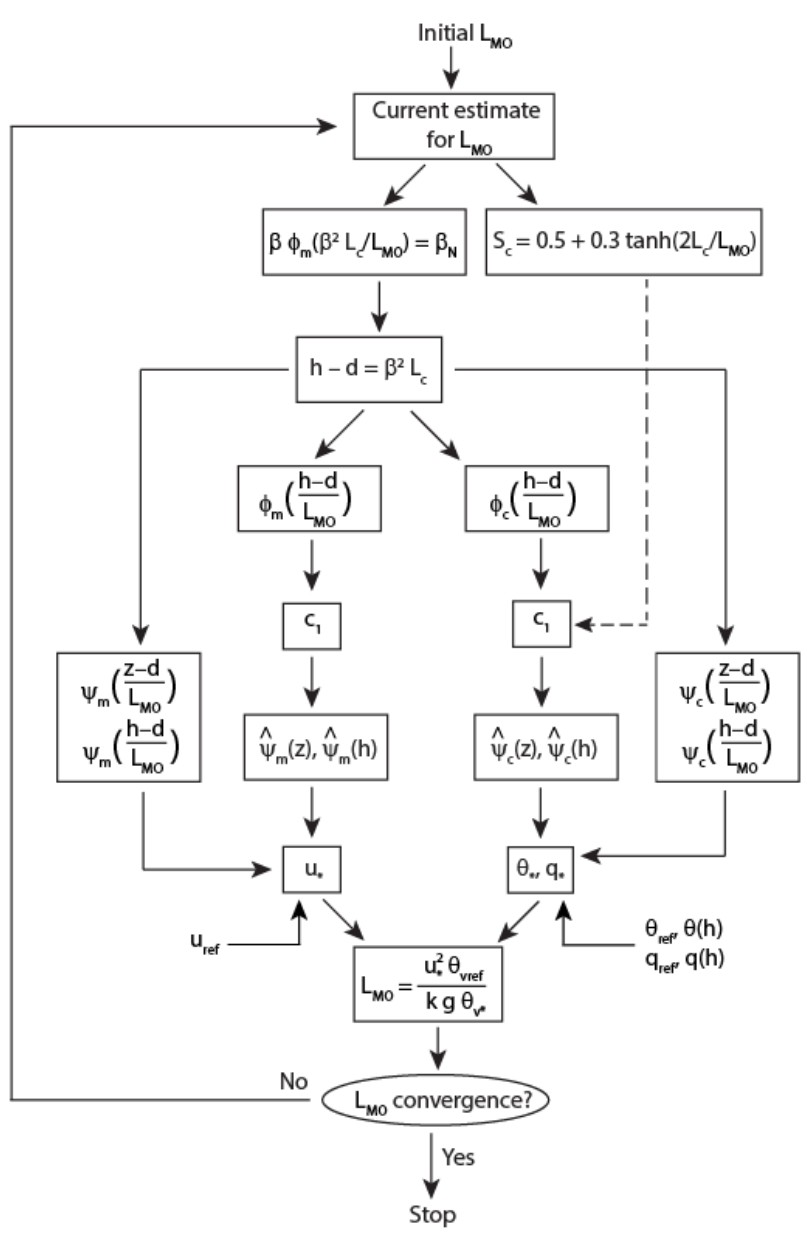


Figure 2. Flow diagram for calculating the Obukhov length ($L_{MO}$).




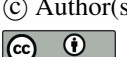


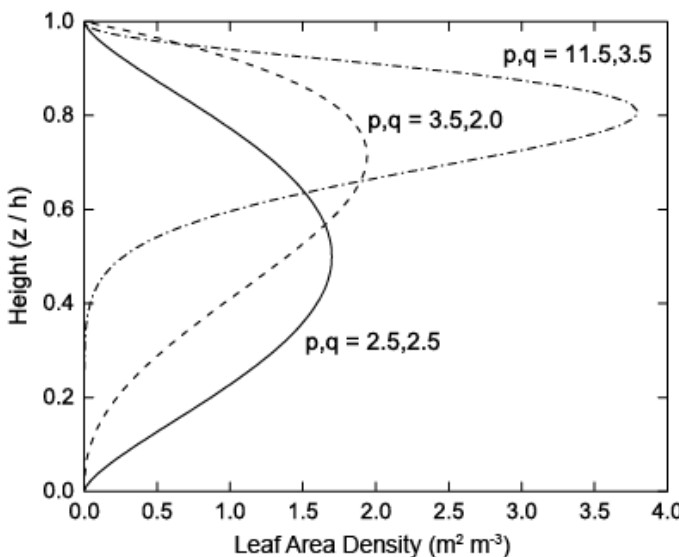


Figure 3. Profiles of leaf area density. Shown are three different canopy profiles for: (i) grass
and crop with $p = q = 2.5$ ; (ii) deciduous and spruce trees with $p = 3.5$ and $q = 2.0$ ; and (iii)
pine trees with $p = 11.5$ and $q = 3.5$ . These profiles are show here with $L_T / h = 0.5$ m$^2$ m$^{-3}$.











Figure 4. Simulations for US-UMB (July 2006). Shown are the average diurnal cycle (GMT) of
sensible heat flux, latent heat flux, friction velocity, radiative temperature, and gross primary
production (GPP) for the observations (blue) and models (red).  The shading denotes ± 1
standard deviation of the random flux error (Richardson et al., 2006, 2012) for $H$ and $\lambda E$ and ±
20% of the mean for GPP and $u_*$. Statistics show sample size ($n$), correlation coefficient ($r$),
slope of the regression line, mean bias, and root mean square error (rmse) between the model and
observations. Left column: CLM4.5. Middle column: ML-RSL. Right column: ML+RSL.











Figure 5. Taylor diagram of net radiation, sensible heat flux, latent heat flux, friction velocity,
radiative temperature, and gross primary production (GPP) for US-UMB. Data points are for the
years 1999–2006 for CLM4.5 (blue) and ML+RSL (red). Simulations are evaluated by the
normalized standard deviation relative to the observations (given by the radial distance of a data
point from the origin) and the correlation with the observations (given by the azimuthal
position). The thick dashed reference line (REF) indicates a normalized standard deviation equal
to one. Model improvement is seen by radial closeness to the REF line and azimuth closeness to
the horizontal axis (correlation coefficient equal to one).





Figure 6. Sensible heat flux in relation to the temperature difference $T_{rad} - T_{ref}$ for US-UMB

(July 2006), US-Me2 (July 2007), and US-ARM (April 2006). Left column: Observations.

Middle column: CLM4.5. Right column: ML+RSL.








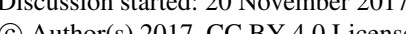

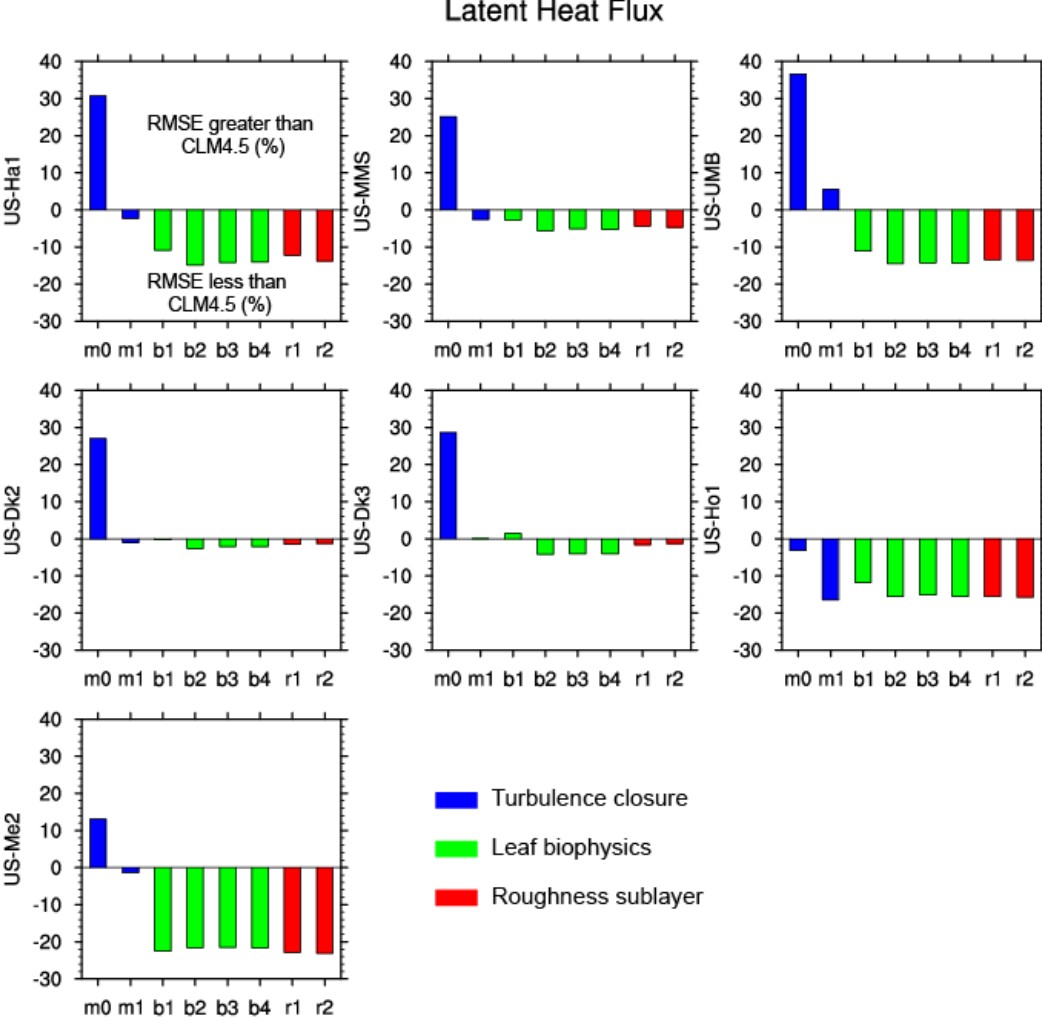


Figure 7. Root mean square error (RMSE) for latent heat flux for the 8 simulations m0–r2.

RMSE for each simulation is given as a percentage of the RMSE for CLM4.5 and averaged

across all years at each of the 7 forest sites. Changes in RMSE between simulations show the

effect of sequentially including new model parameterizations as described in Table 4.







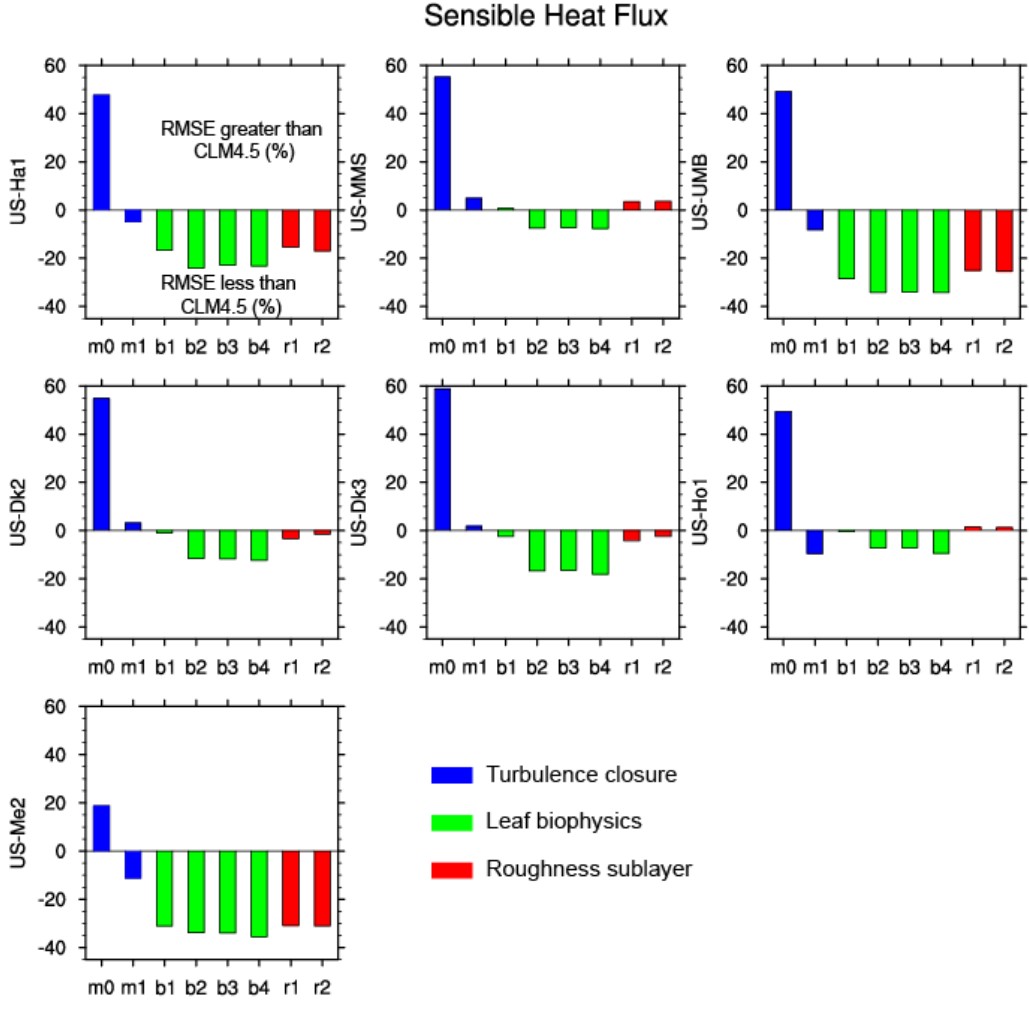


Figure 8. As in Figure 7, but for sensible heat flux.


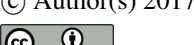



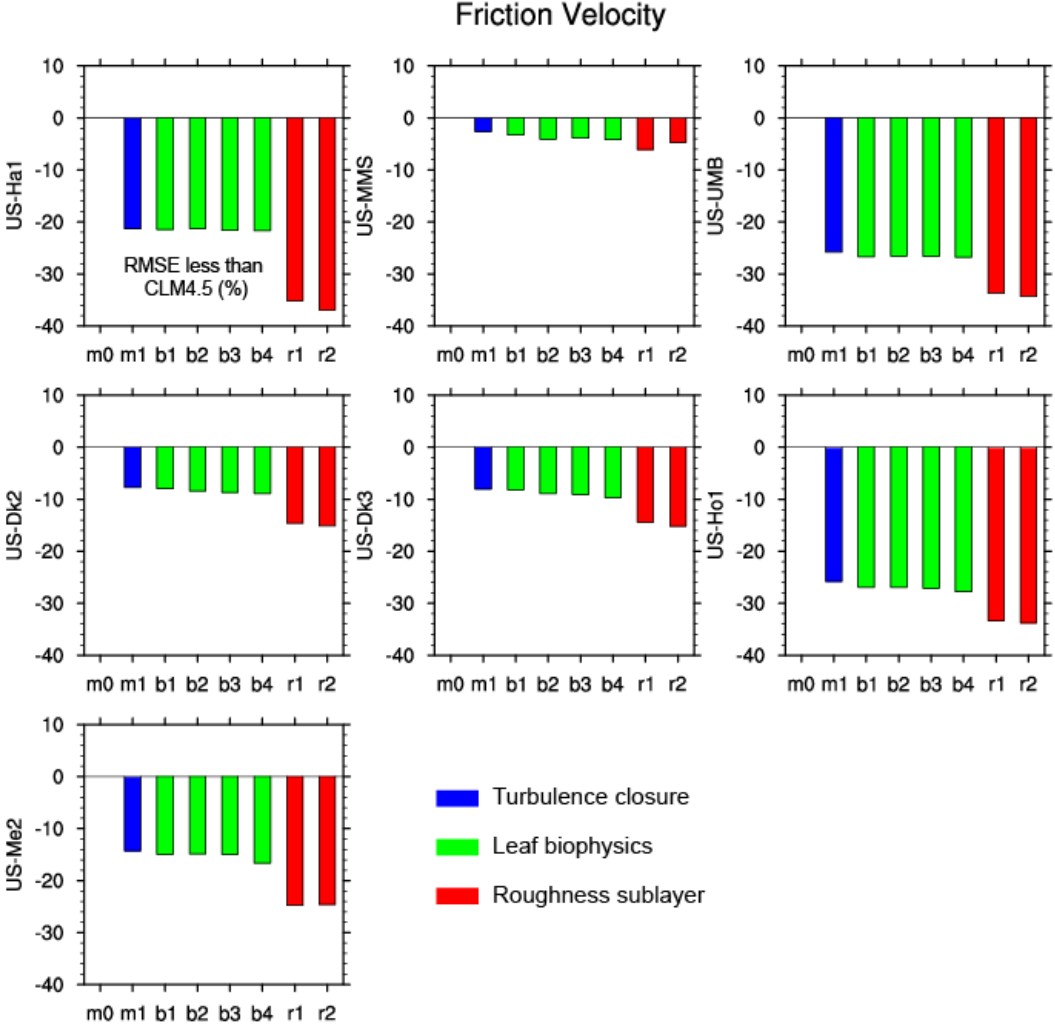


Figure 9. As in Figure 7, but for friction velocity.




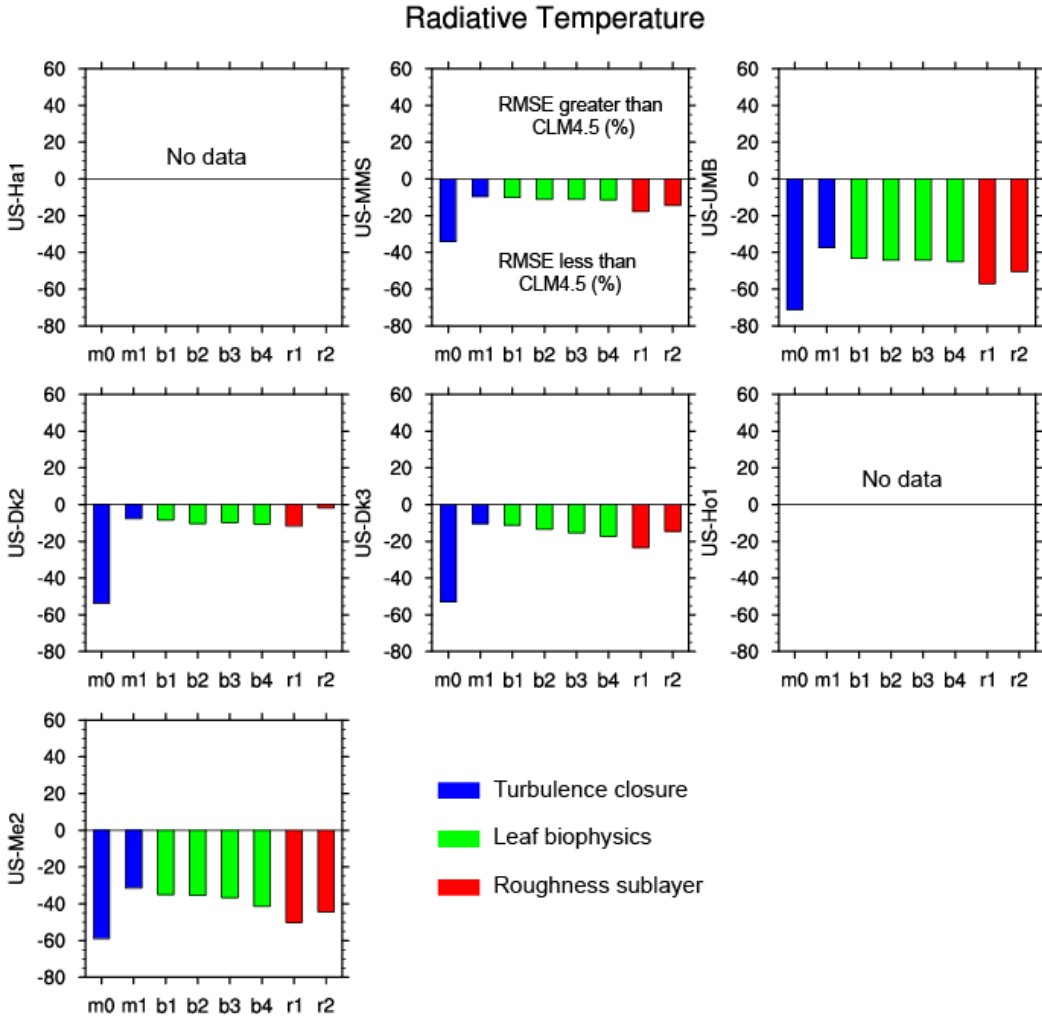


Figure 10. As in Figure 7, but for radiative temperature.








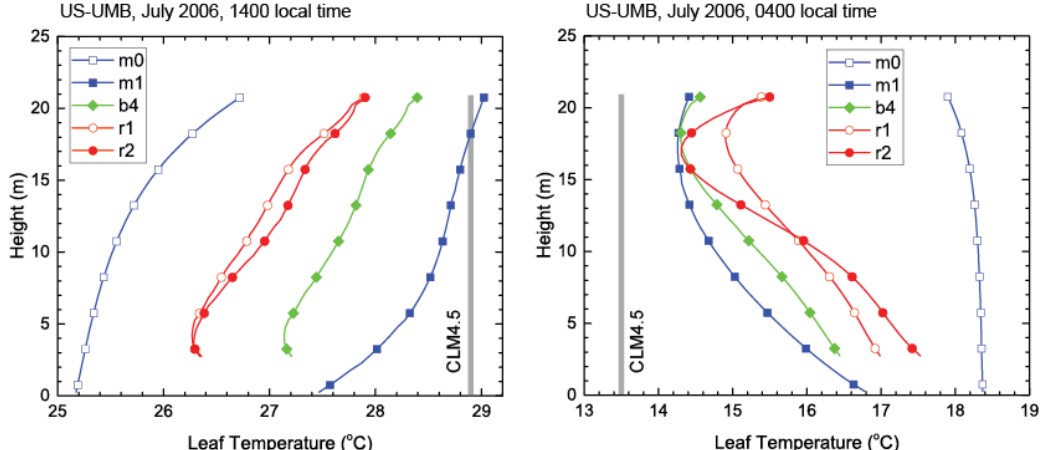


Figure 11. Profiles of leaf temperature for US-UMB averaged for the month of July 2006 at 1400
local time (left panel) and 0400 local time (right panel). Temperature is averaged for sunlit and
shaded leaves at each level in the canopy. Shown are the m0, m1, b4 (ML-RSL), r1, and r2
(ML+RSL) simulations. The CLM4.5 canopy temperature is shown as a thick gray line, but is
not vertically resolved.






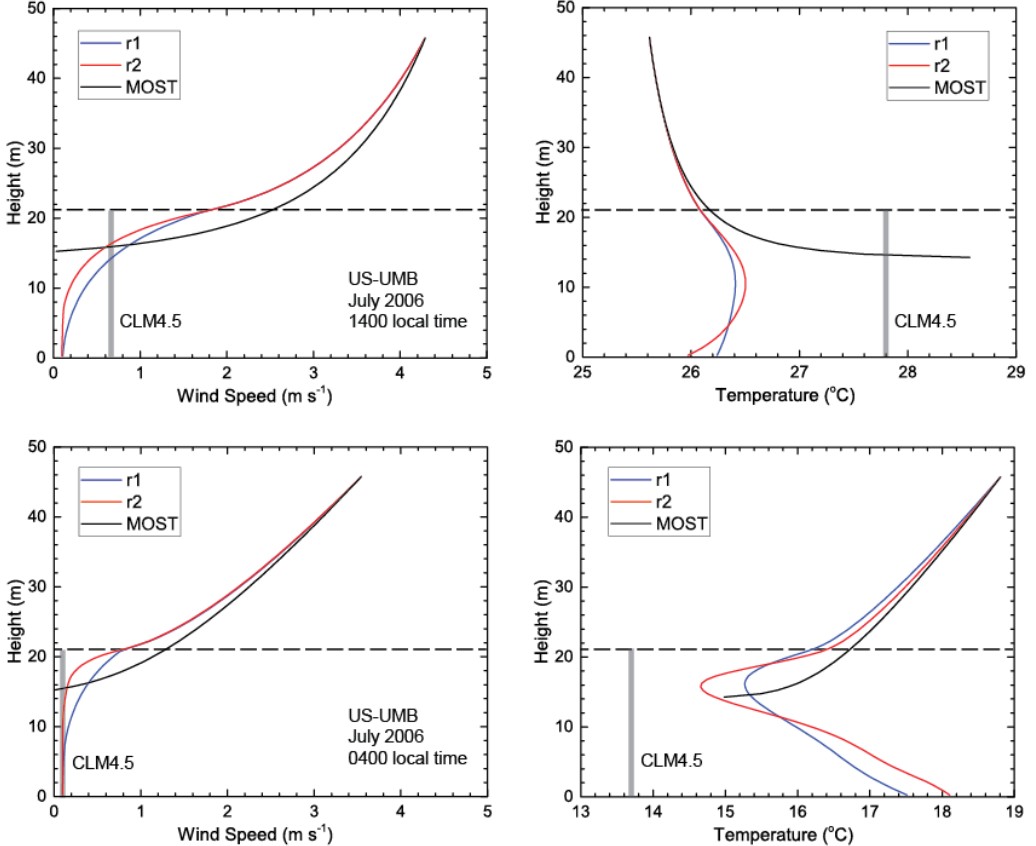


Figure 12. Profiles of wind speed and air temperature for US-UMB (July 2006) at 1400 local
time (top panels) and 0400 local time (bottom panels). Shown are the r1 and r2 simulations
averaged for the month. The dashed line denotes the canopy height. The CLM4.5 canopy wind
speed and air temperature are shown as a thick gray line, but are not vertically resolved. Also
shown are the profiles obtained using MOST extrapolated to the surface. This extrapolation is for
the r2 simulation using Eqs. (19) and (20) but without the RSL and with roughness length and
displacement height specified as in the CLM4.5.