# Peer review of "Modeling canopy-induced turbulence in the Earth system: a unified parameterization of turbulent"

_Geoscientific Model Development, 2017_

## Referee Comment (RC1) · M. Shapkalijevski (Referee) · 21 Dec 2017

*Interactive comments on the manuscript*
*gmd-2017-261 entitled*: Modeling
canopy-induced turbulence in the Earth system:
a unified parameterization of turbulent exchange
within plant canopies and the roughness sublayer
by Bonan et al.

Refferee: Metodija M. Shapkalijevski

December 21, 2017

**General comments**

Improved and more precise evaluation of turbulent exchange of momentum, energy, and passive and active chemical compounds between the land and the atmosphere in presence of vegetation canopy is beneficial for both modeling and measurement communities. This model development report quantifies the canopy and the roughness-sublayer (RSL) induced turbulent effects on surface-atmosphere exchange properties as evaluated by comparing large observational data, Community Land Model version 4.5(CLM4.5) and multi-layer canopy model. The authors concluded that *'the implementation of the RSL improves model performances in terms of sensible heat flux, friction velocity, and radiative temperature, and additional improvement comes from modeling stomatal conductance and canopy physiology beyond what is in the CLM4.5 .'*, which is important and relevant conclusion. The paper is well written and provides the all necessary information of the modeling system.

The main drawback of the paper however, is often not clear separation of the added value of the included RSL parameterization, and the 'Leaf biophysics' incorporation in the model, when presenting and discussing the results (although figures/tables show this clearly). For example, the conclusion sentence, cited above, states that the RSL improves the sensible heat flux, friction velocity and the radiative temperature. This is only true when taken the RSL together with the leaf biophysics improvement in the multi-layer approach, but not entirely true for the sensible and the latent fluxes as seen separately only for the RSL effects (we cannot know this since the RSL here is always linked to the leaf physics of the multi-layer model, and the latter is absent/different in the CLM).

**Specific comments**

**page 2, line 29-30 in the abstract:** please see the same comment in the general note. The effective influence of the RSL on presented quantities would be by comparing the ML-RSL and ML+RSL.

**page 8, Eq. 1, 2, 3, 4:** The fluxes, as stated in the equations, show that they are height dependent (e.g. $\frac{dH}{dz} = f(z)$); but later (**page 15**, Eq. 18-20 are derived from $\frac{dc}{dz} = \frac{c_*}{\kappa z} \Phi_c$ (e.g. Harman and Finnigan 2008, Eq. 12) on the assumption that the fluxes above the canopy are height independent (with $c_* = F_c/\rho u_*$). This seems theoretically incorrect statement and need justification.

**page 9, line 184:** The scalar diffusivity ($K_c$) is assumed to be the same for heat and water vapor. It has to be shown that this is not always the case, especially near the canopy top (e.g. please see Shapkalijevski et al. 2016, Fig. 1)

**page 12, line 242:** *'... additional source fluxes'*, but during day, and sink during night?

**page 17, line 348-349** similar to the comment above on **page 9, line 184:**.

**page 18, line 366** Eq. 27, the roughness length for momentum and scalars are defined as invariant (fixed values), but no reference is given based on what. The RSL theory (Harman and Finnigan 2007; 2008) defines them as variant quantities, dependent on the flow/stratification and canopy properties. Further justification here would be very appreciated

**page 26, line 538:** The wind speed, as simulated including the RSL effects in the flux-gradient relationship of momentum has smaller magnitude compared to the wind speed from the standard MOST. Looking at the profiles provided by Harman and Finnigan (2007), the wind profiles calculated by RSL is generally stronger compared to the wind profiles calculated by MOST. Any comment in the discussion about this would be also very appreciated.

**page 28, line 579** The RSL effects are expected to have larger influence on nocturnal turbulent exchange (as assumed by the theory), due to shear-driven (canopy induced in this case) turbulence dominating over the night (compared to thermal convection during day). This is excellent example that corroborates this assumption.

**page 32, line 666-672** Shapkalijevski et al. (2016) used the RSL theory (Harman and Finnigan 2007; 2008) over a canopy with different sparsity/density and explicitly calculated the $\beta$ and the $l_m/\beta$ scale as function of stability.

**page 66, Figure 1** It could be convenient for the readers if the displacement height and the roughness length are define in the schematic figure.

**References**

Harman, I. N. and Finnigan, J. J.: A simple unified theory for flow in the canopy and roughness sublayer, Boundary-Layer Meteorology, 123, 339363, doi:10.1007, 2007.

Harman, I. N. and Finnigan, J. J.: Scalar Concentration Profiles in the Canopy and Roughness Sublayer, Boundary-Layer Meteorology, 129, 323351, doi:10.1007, 2008.

Shapkalijevski, M., Moene, A. F., Ouwersloot, H. G., Patton, E. G., and Vil-Guerau de Arellano, J.: Influence of canopy seasonal changeson turbulence parameterization within the roughness sublayer over an orchard canopy, Journal of Applied Meteorology and Climatology, pp. 150205, doi:10.1175/JAMC-D-15-0205.1, doi/abs/10.1175/JAMC-D-15-0205.1, 2016.

---

## Referee Comment (RC2) · Anonymous Referee #2 · 23 Jan 2018

"Modeling canopy-induced turbulence in the Earth system: a unified parameterization of turbulent exchange within plant canopies and the roughness sublayer (CLM-ml v0)" by de Bonan *et al.*.

| | |
|---|---|
| Manuscript: | GMD-2017-216 |
| Title: | Modeling canopy-induced turbulence in the Earth system: a unified parameterization of turbulent exchange within plant canopies and the roughness sublayer (CLM-ml v0) |
| Authors: | G.B. Bonan, E.G. Patton, I.N. Harman, K.W. Oleson, J.J. Finnigan, Y. Lu, E.A. Burakowski |

**Recommendation**

Minor revisions

Evaluation of Referee:

| | Excellent | Good | Fair | Poor |
|---|---|---|---|---|
| Scientific significance | x | | | |
| Scientific quality: | x | | | |
| Scientific reproducibility | | x | | |
| Presentation quality | | x | | |

**General**

- The paper describes the canopy-related parameterizations in a land-surface model (LSM) in which the canopy is vertically explicit, various plant-related processes are parameterized rather than kept fixed, and the flow inside the canopy and above the canopy are explicitly coupled in a physically sound way.

- The developed land-surface model nicely brings together a number of relatively recent developments in our understanding of canopy related processes and seeks to couple aspects of canopy processes that often are only studied in isolation.

- The model is tested (offline) on a range of datasets and the testing of the various aspects of the new model is done in a very systematic way, such that (more or less) the relative contribution of each innovation can be quantified.

- The systematic study of the various improvements in the LSM are highly relevant for the scientific community, as well as for the description of land-surface processes in weather and climate models.

- The paper is very well written and clearly structured.

However, I do have some comments:

a. The pedigree of the model being tested is not fully clear, neither is its exact link with CLM4.5. Given the scope of the journal, as well as the importance of a widely used LSM like CLM, it is important to make this crystal clear.

b. Whereas part of the model uses vertically varying plant area densities, the model that describes the in-canopy turbulence profiles and wind profiles assumes a constant plant area density. This inconsistency seems to remain undiscussed.

"Modeling canopy-induced turbulence in the Earth system: a unified parameterization of turbulent exchange within plant canopies and the roughness sublayer (CLM-ml v0)" by de Bonan *et al.*.

c.   Figure 7-10 are to me the core of the analysis, showing how the different model modifications change the skill of the LSM. However, I wonder if the statistic used (RMSE, probably not bias-corrected) is the most informative measure to illustrate and understand the changes in model skill.

d.   Although this is primarily a model-description paper, it does contain a clear research part (which I very much appreciate). However, it would then have been helpful to include a research question that matches the performed research (e.g. 'which of the model modifications had the most important positive impact on model performance for which model output, and for which sites'). Having such a research question would also make the conclusion more concrete.

e.   (partly linked to the previous point) The paper misses a clear synthesis of the model evaluation results: what are the major tendencies with respect to skill: for what type of sites does which type of model improvement (multilayer, plant-physiology or RSL) have an impact on what type of model output. With that synthesis potential users of the model would directly know if the new model would have an important impact on their simulations.

Below I will provide detailed comments

*Note: in the comments below, the comment is preceded by the line number.*

**Detailed comments**

1.   72: in modelling (as opposed to observational studies) the issue is not so much that the flux is larger than inferred from the vertical gradient or difference., but rather the other way around. The lower boundary condition rather acts as a flux boundary condition (at least for daytime conditions) and hence the failure of MOST in describing the flow in the RSL leads to an *overestimation of the vertical differences* (for stable conditions this may be different as the nature of the boundary condition depends on the stability).

2.   75/76: similar remarks as remark 1 hold here: wind speed determines the *link* between temperature/concentration difference and the corresponding flux: it is not necessarily so that the flux is the *dependent variable* as the formulation may suggest.

3.   84-86: the model on which the model that is tested in this paper is based is clearly identified with a reference. However, the relationship of that model to ORCHIDEE and to CLM4.5 is unclear. Please add a clear sketch of the origin of the currently used model, and its relationship to other models mentioned.

4.   94-98: what I miss in the motivation is that with changing profiles of temperature, humidity and wind in the canopy, plant-related processed may also change. Since quite a large part of the simulations in the sensitivity analysis are devoted to those aspects it would be worthwhile to make this link in the introduction.

5.   108: the validation variables are clearly indicated (although I miss an indication of the temporal resolution used: hourly?), but the variables used to force the model are not indicated.

6.   113: I do not see why within a single month soil moisture variations would not need to be accounted for? Is this a rain-free month for each site, is there no dry-down happening? Do I have to interpret this remark as that soil moisture stress of the vegetation is assumed to be absent?

7.   114-130: the enumeration of site parameters seems to be somewhat random (for different sites, different parameters are mentioned). I would suggest to extend the table to include the site-dependent parameters, and to add a reference (as a table footnote) for each site.

8.   144: here CLM4.5 surfaces: if I understand it well (see also remark 3) the multilayer model

> is constructed in such a way that a number of parameterizations are close to what is used in CLM4.5 so that a comparison between CLM4.5 and the simplest version of the multilayer model would –approximately- only test the transition from single layer to multi-layer. As per remark 3: please clarify the strategy.

9. 152: Figure 1: for the reader to appreciate the sensitivities of modelled fluxes to the different step changes in the parameterizations later on, it would perhaps be helpful to sketch a more conceptual picture that shows which variables and which resistances are affected by which part of the model improvements: the multi-layer coupling, the plant-related parameterizations or the turbulence-related (RSL) parameterizations. If not in a separate figure it could perhaps be implemented by using three different colors in Figure 1 to identify which parts are directly affected by which part of the model improvement.

10. Section 2.1.1: for easier reference it would be helpful to introduce an extra level of sectioning: users of your model will more easily be able to find the aspect they need (e.g. (1) canopy-space scalar budget, (2) leave energy balance, (3) vertical discretization, (4) numerical solution).
Since this would make the section numbers excessively long, you could consider to make the model description, which in the end is the main reason for this paper, a separate chapter, rather than section 2.1 (2. Model formulation, 3 Data and methods).

11. 302: the derivation of the RSL-model also requires/implies a *vertically* homogeneous canopy (in terms of leaf area density). In principle this is at odds with the explicit use of vertically varying plant area densities (see section 2.1.3). The effect of this inconsistency seems to remain undiscussed.

12. Chapter 3 would also benefit from a division in subsections.

13. 477 and further (discussion of Table 5). For the interpretation of the results it is important to know what is roughly the partitioning between latent and sensible heat flux: this is an important factor in determining how sensitive fluxes are to changes in the aerodynamic resistance (both between leaf and canopy air, and between canopy and surface layer). The sensitivity to a certain change in aerodynamic resistance may even change sign, and for a given amount of available energy the sensitivities of sensible and latent heat flux are of opposite sign. If the energy partitioning is different between the sites, this might also explain some of the differences in the sensitivities observed in figures 7 to 10.

14. 478: In the interpretation of the results in Table 5 it would be helpful to have an indication as to how significant the change in skill of the new model is, compared to CLM4.5. Some changes are very clear, others seem to be marginal (in both directions). I would suggest to limit the discussion to the significant ones.

15. 499: 'complex': I would say that this type of behavior is well-known: for moderate cooling the turbulence is sustained and a more or less monotonous relationship between sensible heat flux exists. However, when the cooling exceeds a certain limit (or wind speed drops below a certain limit) turbulence vanishes and the relationship between temperature difference (finite) and heat flux (tends to zero) is lost (check literature on 'maximum sustainable heat flux'). In that case the surface temperature is the result of the interplay between radiative cooling, supply of heat flow below (soil of lower canopy) and some remaining weak turbulence that supplies heat from above. It would be interesting to know which of the steps from CLM4.5 to ML+RSL makes the change in realism here (which, by the way, is a very relevant result).

16. 565: if the results are degraded by the inclusion of the RSL description, then apparently the change in flux that resulted from the updated biophysics was too large? Or could there be another reason for this degradation?

17. 573-574: I do not see why the question whether the observations were made inside or above

the RSL would matter here. The relationship between ASL-temperature and canopy temperature is just different between RSL-enabled models and pure MOST. For high canopies the ASL observation is closer (in terms of multiples of RSL height) to the canopy than for low canopies. Hence for high canopies the difference between RSL-estimates of the canopy temperature and MOST-estimate is large as compared to the total vertical temperature difference. On the other hand, for low canopies the largest part of the vertical temperature difference occurs above the RSL (in which MOST is supposed to be valid), hence the error in the within-RSL profile has little weight in the total vertical temperature difference.

18. 701-702: what are these minimum values for the conductances?
19. Figure 4: is the RMSE reported in the figures bias-corrected or does it include the RMSE due to the bias?
20. Figure 5: I wonder why the different years are shown as separate symbols. I would be more interested in seeing all sites plotted in the same figure (with a single symbol giving the multi-year statistics) so that we can try to understand to what extent the different sites show different skills 9as can be seen in table 5).
21. Figure 6: although these figures are very informative, and show a clear change between the different model versions, it is unclear why the points in these figures should be well-behaved. The link between temperature difference and heat flux is indirect: friction velocity and stability are variables that enter into this relationship (or wind speed and stability if one would use a drag-law formulation).
22. Figure 7-10: is the RMSE shown here bias-corrected? If not, it is not full clear whether we look at biases (interesting in themselves, but then show biases in the graphs) or a mix of mean bias and incorrect dynamics.

**Very detailed comments**

1. 87: 'this class' refers to RSL-aware models, or multi-layer models?
2. 106: in Table 1 mean annual temperature and annual (?) precipitation are given. To understand the climatological setting this is OK, but to understand the data that we will be looking at values representative for July might perhaps be more informative.
3. 151-152: if no scalar profiles included in Bonan et al. (2014), then how did the plant-related processes obtain information on in-canopy temperature and humidity?
4. 152: 'The approach': does this refer to the grid?
5. 170: 'vertical flux H': in fact it should be the vertical divergence of the vertical flux that affects the temperature.
6. 179: as in Harman and Finnigan (2007, 2008): it would be useful to refer forward to the location where the parameterization of the turbulent diffusivities is described in this paper.
7. 202-203: you indicate that a conductance is needed for evapotranspiration from partially wetted leaves: please refer forward to equation 12 to reassure the reader that you will take care of this.
8. 214-215: 'The next three terms ...': in fact these three terms describe the flux *divergencei.*
9. 216-217: please refer forward to section 2.1.2 to the description the aerodynamic conductance.
10. 316: also interpret $l_m$ as the mixing length in the canopy. This interpretation now occurs only at line 324.
11. 467: is the modelled upward longwave flux solely determined by the temperature of the upper canopy layer or does the layer below the top also contribute? Not much is said about how radiative transfer is handled (except for the references in line 139).
12. 493: this is a rather long sentence: do you intend to say that these sites where selected

because they had a small RMS for sensible heat flux and surface temperature?
13. 498: 'data': do you mean simulation results or observations?
14. 609-613: check this sentence (long, multiple messages, broken?)
15. Figure 7, line 1225: it is not fully clear what is shown here. I interpret the bar graphs as showing the percentage *change* in RMSE relative to CLM4.5. Then a large negative value would be optimal (-100 would be perfect). In that sense the metric is a bit confusing since showing a mix of positive and negative values might suggest a bias plot to the reader, rather than an RMSE(-change) plot.

---

## Author Comment (AC1) · 20 Feb 2018

**RC1**

We thank the reviewer for his careful reading of the manuscript and attention to detail. These comments, and our responses as described below, improved the focus, clarity, and main points of the manuscript.

General Comments

Improved and more precise evaluation of turbulent exchange of momentum, energy, and passive and active chemical compounds between the land and the atmosphere in presence of vegetation canopy is beneficial for both modeling and measurement communities. This model development report quantifies the canopy and the roughness-sublayer (RSL) induced turbulent effects on surface-atmosphere exchange properties as evaluated by comparing large observational data, Community Land Model version 4.5(CLM4.5) and multi-layer canopy model. The authors concluded that 'the implementation of the RSL improves model performances in terms of sensible heat flux, friction velocity, and radiative temperature, and additional improvement comes from modeling stomatal conductance and canopy physiology beyond what is in the CLM4.5 .', which is important and relevant conclusion. The paper is well written and provides the all necessary information of the modeling system.

The main drawback of the paper however, is often not clear separation of the added value of the included RSL parameterization, and the 'Leaf biophysics' incorporation in the model, when presenting and discussing the results (although figures/tables show this clearly). For example, the conclusion sentence, cited above, states that the RSL improves the sensible heat flux, friction velocity and the radiative temperature. This is only true when taken the RSL together with the leaf biophysics improvement in the multi-layer approach, but not entirely true for the sensible and the latent fluxes as seen separately only for the RSL effects (we cannot know this since the RSL here is always linked to the leaf physics of the multi-layer model, and the latter is absent/different in the CLM).

**Response**: We revised this sentence to distinguish the effects of leaf biophysics from the RSL and further elaborated on this point: "*The multi-layer canopy improves model performance compared to the CLM4.5 in terms of latent and sensible heat fluxes, friction velocity, and radiative temperature. Improvement in latent and sensible heat fluxes comes primarily from advances in modeling stomatal conductance and canopy physiology beyond what is in the CLM4.5. These advances also improve friction velocity and radiative temperature, with additional improvement from the RSL parameterization. The multi-layer canopy combines improvements in both leaf biophysics and canopy-induced turbulence and both contribute to the overall model improvement.*"

page 2, line 29-30 in the abstract: please see the same comment in the general note. The effective influence of the RSL on presented quantities would be by comparing the ML-RSL and ML+RSL.

**Response**: We wrote this sentence to distinguish the effects of leaf biophysics from the RSL: "*Advances in modeling stomatal conductance and canopy physiology beyond what is in the CLM4.5 substantially improve model performance. The signature of the roughness sublayer is most evident in nighttime friction velocity and the diurnal cycle of radiative temperature, but is also seen in sensible heat flux.*"

page 8, Eq. 1, 2, 3, 4: The fluxes, as stated in the equations, show that they are height dependent (e.g. dH/dz=f(z)); but later (page 15, Eq. 18-20 are derived from dc/dz = $c_*$/(kz)$\Phi_c$ (e.g. Harman and Finnigan 2008, Eq. 12) on the assumption that the fuxes above the canopy are height independent (with $c_*=F_c/\rho u_*$). This seems theoretically incorrect statement and need justification.

**Response**: The notation H(z) and E(z) in these equations is for consistency because the equations apply both above and within the canopy. For clarification, we added the sentence: "*Fluxes above the canopy are obtained from MOST flux–gradient relationships as modified for the RSL, and $K_c$ within the canopy is obtained from the momentum and scalar balance equations for plant canopies (section 2.2).*"

page 9, line 184: The scalar diffusivity ($K_c$) is assumed to be the same for heat and water vapor. It has to be shown that this is not always the case, especially near the canopy top (e.g. please see Shapkalijevski et al. 2016, Fig. 1).

**Response**: We revised this to read: "…with $K_c$ the scalar diffusivity ($m^2\,s^{-1}$), assumed to be the same for heat and water vapor *as is common in land surface models though there are exceptions (e.g., Shapkalijevski et al. 2016).*

page 12, line 242: '… additional source fluxes', but during day, and sink during night?

**Response**: We changed "source fluxes" to "*source/sink fluxes*". For consistency, we made the same change to "source flux" throughout the manuscript or deleted "source" as appropriate.

page 17, line 348-349 similar to the comment above on page 9, line 184.

**Response**: See our response to the previous comment.

page 18, line 366 Eq. 27, the roughness length for momentum and scalars are defined as invariant (fixed values), but no reference is given based on what. The RSL theory (Harman and Finnigan 2007; 2008) defines them as variant quantities, dependent on the flow/stratification and canopy properties. Further justification here would be very appreciated

**Response**: The roughness lengths used in Eq. 27 are for the ground surface under the canopy. There are taken from the CLM4.5. We added this reference to the text: "…roughness lengths of the ground for momentum and scalars, respectively, *as in the CLM4.5*…"

page 26, line 538: The wind speed, as simulated including the RSL effects in the flux-gradient relationship of momentum has smaller magnitude compared to the wind speed from the standard MOST. Looking at the profilles provided by Harman and Finnigan (2007), the wind profiles calculated by RSL is generally stronger compared to the wind profiles calculated by MOST. Any comment in the discussion about this would be also very appreciated.

**Response**: The reason for this is that the MOST profiles are calculated using prescribed roughness length and displacement height as in CLM4.5. We note this in the figure caption. The differences in roughness length and displacement height between MOST and RSL change both the value of u(z)/u* at the reference height and also the form of u(z).

page 28, line 579 The RSL effects are expected to have larger influence on nocturnal turbulent exchange (as assumed by the theory), due to shear-driven (canopy induced in this case) turbulence dominating

over the night (compared to thermal convection during day). This is excellent example that corroborates this assumption.

**Response**: We expanded upon this statement as suggested by the reviewer: "…primarily by increasing $u_*$ at night *as expected due to shear-driven turbulence induced by the canopy dominating during night compared with day*."

page 32, line 666-672 Shapkalijevski et al. (2016) used the RSL theory (Harman and Finnigan 2007; 2008) over a canopy with different sparsity/density and explicitly calculated the β and the $l_m$/β scale as function of stability.

**Response**: We added this reference in the introduction: "… *observations above a walnut orchard further support the theory (Shapkalijevski et al. 2016).*"

page 66, Figure 1 It could be convenient for the readers if the displacement height and the roughness length are define in the schematic figure.

**Response**: We added to the figure caption: "*In the CLM4.5, $d$ and $z_0$ are prescribed fractions of canopy height.*"

**RC2**

We thank the reviewer for their careful reading of the manuscript and attention to detail. These comments, and our responses as described below, improved the focus, clarity, and main points of the manuscript.

General comments

a. The pedigree of the model being tested is not fully clear, neither is its exact link with CLM4.5. Given the scope of the journal, as well as the importance of a widely used LSM like CLM, it is important to make this crystal clear.

**Response**: We revised section 2 (Methods) to better clarify the development of the multilayer model and its relation to CLM4.5. First, we split the section into two sections with: "*2 Model description*" and "*3 Model evaluation*". See also our response to detailed comments (10). Second, we revised the first paragraph in the model description to better give the history of the multilayer model and its capability. We show the current model is a further development of the previous work of Bonan et al. (2014). Specifically: "*Here, we describe the formulation of the scalar profiles and the RSL, which were not included in Bonan et al. (2014) and which replace the bulk canopy airspace parameterization.*"

The relationship to ORCHIDEE-CAN (also raised in detailed comments 3) is that we use a similar implicit coupling of the flux-profile equations and numerical solution. We acknowledge their work and point out differences with our own implementation. We revised the text to read: "The implementation is conceptually similar to the implementation of a multi-layer canopy in ORCHIDEE-CAN and that model's implicit numerical coupling of leaf fluxes and scalar profiles (Ryder et al., 2016; Chen et al., 2016). *That numerical scheme is modified here to include sunlit and shaded leaves at each layer in the canopy and also the RSL (Harman and Finnigan 2007, 2008). Whereas ORCHIDEE-CAN uses an implicit calculation of longwave radiative transfer for the leaf energy balance, we retain the Norman (1979) radiative transfer used by Bonan et al. (2014).*"

The relationship to CLM4.5 (also mentioned in detailed comments 3 and 8) is two-fold. First, we clarified the intent of the model simulations. In section 3.2 Model simulations we added: "*The CLM4.5 and the multi-layer canopy differ in several ways (Table 3). To facilitate comparison and to isolate specific model differences, we devised a series of simulations to incrementally test parameterizations changes (Table 4).*" Second, the manuscript is a "Development and technical paper" not a "Model description paper". We are describing a canopy parameterization that can be included in CLM4.5, not a specific version of CLM. We previously mentioned this in the code availability section. We also added a statement in the conclusion: "While this is an advancement over the CLM4.5, much work remains to fully develop this class of model *and to implement the multi-layer canopy parameterization in the CLM*."

b. Whereas part of the model uses vertically varying plant area densities, the model that describes the in-canopy turbulence profiles and wind profiles assumes a constant plant area density. This inconsistency seems to remain undiscussed.

**Response**: We added a sentence to our discussion of advantages and limitations in the model: "*The canopy length scale $L_c$ is assumed to be constant with height as in Eq. (56) and is thought to be more conservative than either leaf area density or the leaf drag coefficient separately (Harman and Finnigan (2007). Massman (1997) developed a first-order closure canopy turbulence parameterization that accounts for vertical variation in leaf area density, but that is not considered here.*"

c. Figure 7-10 are to me the core of the analysis, showing how the different model modifications change the skill of the LSM. However, I wonder if the statistic used (RMSE, probably not bias-corrected) is the most informative measure to illustrate and understand the changes in model skill.

**Response**: We used RMSE because it is the most easily interpreted metric of model performance and improvement by sequentially adding new parameterizations. It is not biased corrected and simply assesses the summed error between the model and observations. We also looked at the Taylor skill score as a metric of model performance. It gives a similar assessment of the individual parameterization changes, but the reviewer noted in detailed comment (14) the difficulty in interpreting differences in model skill scores.

d. Although this is primarily a model-description paper, it does contain a clear research part (which I very much appreciate). However, it would then have been helpful to include a research question that matches the performed research (e.g. 'which of the model modifications had the most important positive impact on model performance for which model output, and for which sites'). Having such a research question would also make the conclusion more concrete.

**Response**: We added to the last paragraph of the introduction: "*The previous model development of Bonan et al. (2014) included improvements to stomatal conductance and canopy physiology compared with the CLM4.5. We contrast those developments with the RSL parameterization described herein and compare tall forest with short herbaceous vegetation to ascertain which aspects of the multi-layer canopy most improve the model.*"

e. (partly linked to the previous point) The paper misses a clear synthesis of the model evaluation results: what are the major tendencies with respect to skill: for what type of sites does which type of model improvement (multilayer, plant-physiology or RSL) have an impact on what type of model output. With that synthesis potential users of the model would directly know if the new model would have an important impact on their simulations.

**Response**: The biggest difference we see between sites relates to forest versus herbaceous. The plant physiological improvements occur across sites, but the RSL improvements most consistently occur at forest sites. We revised the abstract to read:

"*Advances in modeling stomatal conductance and canopy physiology beyond what is in the CLM4.5 substantially improve model performance at the forest sites. The signature of the roughness sublayer is most evident in nighttime friction velocity and the diurnal cycle of radiative temperature, but is also seen in sensible heat flux.*"

and:

"*The herbaceous sites also show model improvements, but the improvements are related less systematically to the roughness sublayer parameterization in these canopies.*"

Detailed comments

1. 72: in modelling (as opposed to observational studies) the issue is not so much that the flux is larger than inferred from the vertical gradient or difference, but rather the other way around. The lower boundary condition rather acts as a flux boundary condition (at least for daytime conditions) and hence the failure of MOST in describing the flow in the RSL leads to an *overestimation of the vertical differences* (for stable conditions this may be different as the nature of the boundary condition depends on the stability).

**Response**: We changed the wording to "… within the RSL *flux–profile relationships differ from MOST.*"

2. 75/76: similar remarks as remark 1 hold here: wind speed determines the *link* between temperature/concentration difference and the corresponding flux: it is not necessarily so that the flux is the *dependent variable* as the formulation may suggest.

**Response**: Our intent with this sentence is to show that land surface models must parameterize within-canopy turbulent processes in some manner. We changed the text to read: "*Dual-source land surface models also require parameterization of turbulent processes within the canopy. Following BATS (Dickinson et al., 1986), the CLM4.5 uses an ad-hoc parameterization without explicitly representing turbulence.*"

3. 84-86: the model on which the model that is tested in this paper is based is clearly identified with a reference. However, the relationship of that model to ORCHIDEE and to CLM4.5 is unclear. Please add a clear sketch of the origin of the currently used model, and its relationship to other models mentioned.

**Response**: The particular reference to ORCHIDEE here is merely to acknowledge the previous work of the ORCHIDEE group to develop a multi-layer version of their model (also published in GMD). The specific lineage of that to our work is discussed later. See our response to general comments (a).

4. 94-98: what I miss in the motivation is that with changing profiles of temperature, humidity and wind in the canopy, plant-related processed may also change. Since quite a large part of the simulations in the sensitivity analysis are devoted to those aspects it would be worthwhile to make this link in the introduction.

**Response**: We added to the introduction: "*We show that the resulting within-canopy profiles of temperature, humidity, and wind speed are a crucial aspect of the leaf to canopy flux scaling.*"

5. 108: the validation variables are clearly indicated (although I miss an indication of the temporal resolution used: hourly?), but the variables used to force the model are not indicated.

**Response**: The resolution is 30 minutes or 60 minutes depending on tower site. We added the sentence: "*The tower forcing and fluxes have a resolution of 30 minutes except for four sites (US-Ha1, US-MMS, US-UMB, US-Ne3) with 60 minute resolution.*" We also added text for the forcing variables: "*downwelling solar and longwave radiation, air temperature, relative humidity, wind speed, surface pressure, precipitation, and tower height*".

6. 113: I do not see why within a single month soil moisture variations would not need to be accounted for? Is this a rain-free month for each site, is there no dry-down happening? Do I have to interpret this remark as that soil moisture stress of the vegetation is assumed to be absent?

**Response**: We want to evaluate the canopy physics parameterizations in a clean manner without confounding effects from large changes in soil moisture. We revised the sentence to read: "… *so as to evaluate the canopy physics parameterizations without confounding effects of seasonal changes in soil water.*"

7. 114-130: the enumeration of site parameters seems to be somewhat random (for different sites, different parameters are mentioned). I would suggest to extend the table to include the site-dependent parameters, and to add a reference (as a table footnote) for each site.

**Response**: The text in these two paragraphs (now moved to *section 3.1 Flux tower data*) describes how we obtained vegetation data for the new sites not previously described in Bonan et al. (2014). This includes the type of crop (for the agricultural sites), canopy height, and leaf area index. These variables are provided in Table 2, and the text here documents how those values were obtained. Also, we provide documentation for the tower data.

8. 144: here CLM4.5 surfaces: if I understand it well (see also remark 3) the multilayer model is constructed in such a way that a number of parameterizations are close to what is used in CLM4.5 so that a comparison between CLM4.5 and the simplest version of the multilayer model would – approximately- only test the transition from single layer to multi-layer. As per remark 3: please clarify the strategy.

**Response**: See our response to general comments (a).

9. 152: Figure 1: for the reader to appreciate the sensitivities of modelled fluxes to the different step changes in the parameterizations later on, it would perhaps be helpful to sketch a more conceptual picture that shows which variables and which resistances are affected by which part of the model improvements: the multi-layer coupling, the plant-related parameterizations or the turbulence-related (RSL) parameterizations. If not in a separate figure it could perhaps be implemented by using three different colors in Figure 1 to identify which parts are directly affected by which part of the model improvement.

**Response**: The intent of Figure 1 is to show readers the numerical grid used in the multilayer canopy and contrast this representation with the dual source canopy used in the CLM4.5. Table 3 specifically describes parameterization differences between the multilayer canopy and the CLM4.5.

10. Section 2.1.1: for easier reference it would be helpful to introduce an extra level of sectioning: users of your model will more easily be able to find the aspect they need (e.g. (1) canopy-space scalar budget, (2) leave energy balance, (3) vertical discretization, (4) numerical solution). Since this would make the section numbers excessively long, you could consider to make the model description, which in the end is the main reason for this paper, a separate chapter, rather than section 2.1 (2. Model formulation, 3 Data and methods).

**Response**: We split section 2 (Methods) into two separate sections: *2 Model description* and *3 Model evaluation*. The multilayer canopy has two main components: (1) the canopy flux-profile equations and (2) the roughness sublayer parameterization. These are described in separate sub-sections of the model

description. Additional sub-sections describe leaf area density and leaf heat capacity. We note that section 2.1 (flux-profile equations) follows the sequence that the reviewer suggested.

11. 302: the derivation of the RSL-model also requires/implies a *vertically* homogeneous canopy (in terms of leaf area density). In principle this is at odds with the explicit use of vertically varying plant area densities (see section 2.1.3). The effect of this inconsistency seems to remain undiscussed.

**Response**: See our response to general comments (b).

12. Chapter 3 would also benefit from a division in subsections.

**Response**: We divided the results into three sub-sections: *4.1 Model evaluation*, *4.2 Effect of specific parameterizations*, and *4.3 Canopy profiles*

13. 477 and further (discussion of Table 5). For the interpretation of the results it is important to know what is roughly the partitioning between latent and sensible heat flux: this is an important factor in determining how sensitive fluxes are to changes in the aerodynamic resistance (both between leaf and canopy air, and between canopy and surface layer). The sensitivity to a certain change in aerodynamic resistance may even change sign, and for a given amount of available energy the sensitivities of sensible and latent heat flux are of opposite sign. If the energy partitioning is different between the sites, this might also explain some of the differences in the sensitivities observed in figures 7 to 10.

**Response**: We agree that this is a useful point, but it is beyond the scope of the manuscript. There is a rich literature considering the differences between aerodynamic and physiological controls of surface fluxes and when, for example, a change in stomatal resistance or aerodynamic resistance may or may not affect latent heat flux. The manuscript is already quite long, and to address this beyond a cursory manner would require substantial text and figures. We prefer that the manuscript remain focused on our intent: that the CLM4.5 canopy parameterization is flawed, and that a multilayer canopy model with improved leaf biophysics and turbulence improves upon the CLM4.5.

14. 478: In the interpretation of the results in Table 5 it would be helpful to have an indication as to how significant the change in skill of the new model is, compared to CLM4.5. Some changes are very clear, others seem to be marginal (in both directions). I would suggest to limit the discussion to the significant ones.

**Response**: We use Table 5 as a summary to assess whether the model is, overall across many sites, performing better than CLM4.5. We then use Figures 4, 5, and 6 as detailed flux evaluations for specific sites. Figures 7-10 address why the model is performing better. We agree that the interpretation of the Taylor skill score is not necessarily intuitive (how much better is 0.94 vs. 0.92; US-Ha1 latent heat flux). Nonetheless, the skill score is a composite measure of the data points on a Taylor plot such as Figure 5. The average skill scores presented in Table 5 are exactly what the reviewer requested in comment (20) below.

15. 499: 'complex': I would say that this type of behavior is well-known: for moderate cooling the turbulence is sustained and a more or less monotonous relationship between sensible heat flux exists. However, when the cooling exceeds a certain limit (or wind speed drops below a certain limit) turbulence vanishes and the relationship between temperature difference (finite) and heat flux (tends to zero) is lost (check literature on 'maximum sustainable heat flux'). In that case the surface

temperature is the result of the interplay between radiative cooling, supply of heat flow below (soil of lower canopy) and some remaining weak turbulence that supplies heat from above. It would be interesting to know which of the steps from CLM4.5 to ML+RSL makes the change in realism here (which, by the way, is a very relevant result).

**Response**: The functional relationship between H and ΔT is an important way to test the model in addition to direct comparison between observed and modeled fluxes such as presented in Figure 4. The main point here is that CLM4.5 shows a very different pattern compared to the observations (for the forest sites) and that the new canopy model better matches the observations. We revised the figure to include the ML-RSL simulation so that we can clearly distinguish the influence of the RSL. We added a sentence to the results section: "*The primary effect of the RSL is to reduce high daytime temperatures and to increase sensible heat transfer to the surface at night*"; and to the discussion of Figure 6: "*Additional improvement, as expected from the RSL theory, is seen during moderately stable periods, which in turn reduces surface cooling.*"

16. 565: if the results are degraded by the inclusion of the RSL description, then apparently the change in flux that resulted from the updated biophysics was too large? Or could there be another reason for this degradation?

**Response**: The main point here is that the RSL parameterization cannot be evaluated independent of changes in leaf biophysics. We return to this point later in the conclusions.

17. 573-574: I do not see why the question whether the observations were made inside or above the RSL would matter here. The relationship between ASL-temperature and canopy temperature is just different between RSL-enabled models and pure MOST. For high canopies the ASL observation is closer (in terms of multiples of RSL height) to the canopy than for low canopies. Hence for high canopies the difference between RSL-estimates of the canopy temperature and MOST-estimate is large as compared to the total vertical temperature difference. On the other hand, for low canopies the largest part of the vertical temperature difference occurs above the RSL (in which MOST is supposed to be valid), hence the error in the within-RSL profile has little weight in the total vertical temperature difference.

**Response**: We deleted the phrase "because the measurements were taken above the RSL."

18. 701-702: what are these minimum values for the conductances?

**Response**: The maximum resistance is 500 s/m. See our discussion of Eqs. 24-27, which describe the conductances.

19. Figure 4: is the RMSE reported in the figures bias-corrected or does it include the RMSE due to the bias?

**Response**: The RMSE is not bias corrected. See our response to general comment (c).

20. Figure 5: I wonder why the different years are shown as separate symbols. I would be more interested in seeing all sites plotted in the same figure (with a single symbol giving the multi-year statistics) so that we can try to understand to what extent the different sites show different skills as can be seen in table 5).

**Response**: Table 5 conveys the information on model performance at each site (averaged across years) that the reviewer requests. Figure 4 then provides a detailed analysis of the diurnal cycle at one site for one year (US-UMB) and Figure 5 provides an analysis of all years at that site.

21. Figure 6: although these figures are very informative, and show a clear change between the different model versions, it is unclear why the points in these figures should be well behaved. The link between temperature difference and heat flux is indirect: friction velocity and stability are variables that enter into this relationship (or wind speed and stability if one would use a drag-law formulation).

**Response**: See comment (15) above.

22. Figure 7-10: is the RMSE shown here bias-corrected? If not, it is not full clear whether we look at biases (interesting in themselves, but then show biases in the graphs) or a mix of mean bias and incorrect dynamics.

**Response**: The RMSE is not bias corrected. See our response to general comment (c).

Very detailed comments

1. 87: 'this class' refers to RSL-aware models, or multi-layer models?

**Response**: We changed "this class of canopy models" to "*multilayer models*".

2. 106: in Table 1 mean annual temperature and annual (?) precipitation are given. To understand the climatological setting this is OK, but to understand the data that we will be looking at values representative for July might perhaps be more informative.

**Response**: We updated the table to provide values for the particular month.

3. 151-152: if no scalar profiles included in Bonan et al. (2014), then how did the plant-related processes obtain information on in-canopy temperature and humidity?

**Response**: The model of Bonan et al. (2014) used a bulk canopy air space parameterization. We added text to state this: "*Temperature, humidity, and wind speed in the canopy are calculated using a bulk canopy airspace.*" We also added text to explain that the new model replaces the bulk canopy airspace: "Here, we describe the formulation of the scalar profiles and the RSL, which were not included in Bonan et al. (2014) *and which replace the bulk canopy airspace parameterization*."

4. 152: 'The approach': does this refer to the grid?

**Response**: We changed "approach" to "*implementation*".

5. 170: 'vertical flux H': in fact it should be the vertical divergence of the vertical flux that affects the temperature.

**Response**: We changed this to read vertical flux divergence of H. We made the same change for E in the next equation.

6. 179: as in Harman and Finnigan (2007, 2008): it would be useful to refer forward to the location where the parameterization of the turbulent diffusivities is described in this paper.

**Response**: We modified the text to refer to section 2.2, where this is discussed.

7. 202-203: you indicate that a conductance is needed for evapotranspiration from partially wetted leaves: please refer forward to equation 12 to reassure the reader that you will take care of this.

**Response**: We modified the text to refer to equation 12.

8. 214-215: 'The next three terms ...': in fact these three terms describe the flux *divergence*.

**Response**: Corrected.

9. 216-217: please refer forward to section 2.1.2 to the description the aerodynamic conductance.

**Response**: We added text to refer to equations 24 and 26.

10. 316: also interpret $l_m$ as the mixing length in the canopy. This interpretation now occurs only at line 324.

**Response**: We are specifically referring to $l_m/\beta$ not $l_m$. We clarified this by changing "This length scale is…" to "The length scale $l_m/\beta$ is…"

11. 467: is the modelled upward longwave flux solely determined by the temperature of the upper canopy layer or does the layer below the top also contribute? Not much is said about how radiative transfer is handled (except for the references in line 139).

**Response**: Longwave radiation is treated in a multilayer framework so that all layers contribute to the upward flux above the canopy. We refer the reviewer (and readers of the manuscript) to the Norman (1979) paper that we cite.

12. 493: this is a rather long sentence: do you intend to say that these sites where selected because they had a small RMS for sensible heat flux and surface temperature?

**Response**: Yes. We broke the sentence into two parts and explained that the sites were selected because they have small RMSE: "*These sites were chosen because the root mean square error of the model (ML+RSL) is low for  H and $T_{rad}$.*"

13. 498: 'data': do you mean simulation results or observations?

**Response**: We changed "data" to "*CLM4.5 data*".

14. 609-613: check this sentence (long, multiple messages, broken?)

**Response**: We broke this into separate sentences: "*The importance of within-canopy temperature gradients is seen in forest canopies. The microclimatic influence of dense forest canopies buffers the impact of macroclimatic warming on understory plants (De Frenne et al., 2013), and the vertical climatic gradients in tropical rainforests are steeper than elevation or latitudinal gradients (Scheffers et al., 2013).*"

15. Figure 7, line 1225: it is not fully clear what is shown here. I interpret the bar graphs as showing the percentage *change* in RMSE relative to CLM4.5. Then a large negative value would be optimal (-100 would be perfect). In that sense the metric is a bit confusing since showing a mix of positive and negative values might suggest a bias plot to the reader, rather than an RMSE(-change) plot.

**Response**: The reviewer is correct in their interpretation of the bar graphs. We are showing the reduction in RMSE relative to CLM4.5. We added text to the figure caption: "*A negative value shows a reduction in RMSE relative to CLM4.5 and indicates model improvement.*"